# Di- and Triselenoesters—Promising Drug Candidates for the Future Therapy of Triple-Negative Breast Cancer

**DOI:** 10.3390/ijms25147764

**Published:** 2024-07-16

**Authors:** Dominika Radomska, Robert Czarnomysy, Anna Szymanowska, Dominik Radomski, Magda Chalecka, Arkadiusz Surazynski, Enrique Domínguez-Álvarez, Anna Bielawska, Krzysztof Bielawski

**Affiliations:** 1Department of Synthesis and Technology of Drugs, Medical University of Bialystok, Kilinskiego 1, 15-089 Bialystok, Poland; 2Department of Experimental Therapeutics, The University of Texas MD Anderson Cancer Center, Houston, TX 77054, USA; 3Department of Medicinal Chemistry, Medical University of Bialystok, Mickiewicza 2D, 15-222 Bialystok, Poland; 4Instituto de Química Orgánica General (IQOG-CSIC), Consejo Superior de Investigaciones Científicas, Juan de la Cierva 3, 28006 Madrid, Spain; 5Department of Biotechnology, Medical University of Bialystok, Kilinskiego 1, 15-089 Bialystok, Poland; anna.bielawska@umb.edu.pl

**Keywords:** breast cancer, triple-negative breast cancer, cancer treatment, anticancer drugs, selenium compounds, organoselenium compounds, selenoesters, apoptosis, autophagy, signaling pathways

## Abstract

Breast cancer is a major malignancy among women, characterized by a high mortality rate. The available literature evidence indicates that selenium, as a trace element, has chemopreventive properties against many types of cancer; as such, compounds containing it in their structure may potentially exhibit anticancer activity. Accordingly, we have undertaken a study to evaluate the effects of novel selenoesters (EDAG-1, -7, -8, -10) on MCF-7 and MDA-MB-231 breast cancer cells. Our analysis included investigations of cell proliferation and viability as well as cytometric determinations of apoptosis/autophagy induction, changes in mitochondrial membrane polarity (ΔΨ_m_), caspase 3/7, 8, and 9 activities, and Bax, Bcl-2, p53, Akt, AMPK, and LC3A/B proteins. The obtained data revealed that the tested derivatives are highly cytotoxic and inhibit cell proliferation even at nanomolar doses (0.41–0.79 µM). Importantly, their strong proapoptotic properties (↑ caspase 3/7) are attributable to the effects on both the extrinsic (↑ caspase 8) and intrinsic (↓ ΔΨ_m_ and Bcl-2, ↑ Bax, p53, and caspase 9) pathways of apoptosis. Moreover, the tested compounds are autophagy activators (↓ Akt, ↑ autophagosomes and autolysosomes, AMPK, LC3A/B). In summary, the potent anticancer activity suggests that the tested compounds may be promising drug candidates for future breast cancer therapy.

## 1. Introduction

The female population has the highest incidence of breast cancer (BC). In the United States (US) alone, the incidence rate of this type of cancer has been steadily increasing by about 0.5% year-on-year since 2000. In the US, it is estimated that the number of new diagnoses and deaths related to BC in 2023 was 300,590 and 43,700, respectively [1]. These figures exceed the recorded 279,100 cases and 42,690 deaths in 2020. Increasing chemoresistance, resulting in high mortality in this patient population, has been a significant clinical problem challenging researchers worldwide for many years [1,2]. Therefore, the search for new, more effective methods of detecting cancerous lesions at their early stages and creating drugs with potent anticancer activity, including those overcoming multidrug resistance (MDR) and targeting specific molecular targets, is ongoing all the time.

For almost 50 years, cisplatin has been widely used in oncology treatment, including in BC therapy. Unfortunately, this drug causes many severe side effects (including nephro-, oto-, and neurotoxicity), which in many clinical cases significantly limits the use of its effective dose. In addition, cancer cells quickly acquire adaptive mechanisms under the influence of this substance, developing resistance [3,4]. It has been proven that the initial therapeutic response to cisplatin is satisfactory; however, during long-term therapy, its effectiveness declines and the disease recurs [5]. Thus, both side effects and resistance to this drug prompt the design and search for substances with high anticancer activity characterized by a favorable therapeutic index and the ability to overcome MDR.

Selenium (Se), despite being a trace element in the human body, plays many crucial roles that are essential in maintaining homeostasis. It has been shown that in addition to its main function—antioxidant activity—it also shows a chemopreventive effect against many cancers, including breast cancer. Although the exact mechanism of how it works is not fully understood, the available evidence indicates that this element, among other things, causes cell cycle arrest, an increase in p53 protein expression, and induction of apoptosis, as well as the suppression of neoangiogenesis and cell proliferation [6,7].

Selenoesters are compounds with the general formula R_1_-CO-Se-R_2_, which undergo rapid intracellular hydrolysis or enzymatic reduction (via the enzymes carbonic anhydrase or acetylcholinesterase) to Se-forms exhibiting anticancer activity [8]. However, their mechanism of action in cancer cells is not fully understood. To date, it has been established that derivatives belonging to this class of Se compounds show potent cytotoxic activity against many cancers, including colon and lung [9], and also exert the ability to overcome resistance [10,11]. These properties appear to be very promising, especially in the context of the occurrence of MDR in cancer cells, prompting us to undertake further research in this area. In our previous work [12], we evaluated the anticancer effect of compounds containing one selenoester group (EDA-71, E-NS-4, Figure 1) against the human breast cancer cells MCF-7 and MDA-MB-231.

This time, we wanted to analyze the impact of a higher number of Se atoms in the molecule on the activity of the tested compounds toward malignant cells and to expand our knowledge of their molecular mechanism of action. Hence, our research has focused on thoroughly assessing the anticancer potential of the novel selenoesters (EDAG-1, EDAG-7, EDAG-8, and EDAG-10) in MCF-7 estrogen-dependent breast cancer and MDA-MB-231 triple-negative breast cancer cells.

## 2. Results

### 2.1. EDAG-1, -7, -8, and -10 as Highly Cytotoxic Compounds and Effective Inhibitors of Breast Cancer Cell Proliferation

To investigate the cytotoxic effects of the tested compounds (EDAG-1, -7, -8, and -10; Figure 2) against breast cancer cells (MCF-7 and MDA-MB-231) and MCF-10A normal human breast epithelial cells, an MTT assay was performed after 24 h of treatment at varying concentrations (0–5 µM). The study utilized cisplatin as the reference compound. All selenoesters exhibited potent toxic effects against the tested cell lines (Figure 3), especially derivatives with a ketone end fragment. The EDAG-1 was the most active against MCF-7 cells, followed by EDAG-8 and EDAG-10, while EDAG-7 exhibited the weakest activity. Meanwhile, the highest efficacy against MDA-MB-231 triple-negative breast cancer cells was observed for the compound EDAG-8. Another equally active compound against this cell line was EDAG-1, while EDAG-10 was almost 4-fold weaker, and the IC_50_ for EDAG-7 was greater than the range of tested concentrations (IC_50_ > 5 µM). In the case of MCF-10A normal breast epithelial cells, derivatives containing a ketone end fragment (EDAG-1 and -8) were more toxic than the compounds containing a nitrile end group in their structure (EDAG-7 and -10). However, cisplatin did not exhibit such potent cytotoxic activity, and the IC_50_ values for all the investigated cell lines were greater than the range of the tested concentrations (>5.0 µM; Figure 3).

The two selenoesters with the highest cytotoxic activity, namely EDAG-1 and EDAG-8 (Figure 2), were chosen for detailed research into the molecular mechanism of their anticancer effects in breast cancer cells.

In the next investigation, we decided to evaluate the impact of novel selenoesters (EDAG-1, -7, -8, and -10) on DNA biosynthesis. For this purpose, MCF-7 and MDA-MB-231 breast cancer cells and MCF-10A human normal breast epithelial cells were exposed for 24 h to these compounds at varied concentrations (0–5 µM; Figure 4). The study utilized cisplatin as the reference compound. As in the viability assay (Figure 3), the most active compounds were EDAG-1 and EDAG-8 (Figure 2). Moreover, it is noteworthy that the derivative EDAG-8 exhibited the most potent inhibitory effect on DNA biosynthesis in triple-negative breast cancer cells (IC_50_ = 0.47 ± 0.08 µM). The remaining two selenoesters (EDAG-7 and EDAG-10; Figure 2) showed 2–3-fold weaker antiproliferative activity than the most active derivatives. In turn, cisplatin exhibited insignificant effects on the DNA biosynthesis process in the tested cell lines, and no IC_50_ value was reached in the used concentration range (>5.0 µM; Figure 4).

### 2.2. EDAG-1 and EDAG-8 as Potent Inducers of Apoptosis Occurring via the Receptor and Mitochondrial Pathways in Breast Cancer Cells

Current trends in the design and synthesis of compounds with anticancer activity are mainly focused on creating drugs that can induce the process of apoptosis in cancer cells, leading to cancer cell death [13,14]. Therefore, we evaluated the proapoptotic activity of the most active tested compounds (EDAG-1 and EDAG-8) and cisplatin at two concentrations (0.5 and 1 µM) in MCF-7 and MDA-MB-231 breast cancer cells and MCF-10A human normal breast epithelial cells after 24 h of exposure. In the beginning, it should be noted that both selenoesters triggered the apoptotic process, in which the late apoptotic cell population predominated. EDAG-8 proved to be the most active compound (Figure 5). Similar values of the apoptotic cell population compared to EDAG-8 at a concentration of 0.5 µM were noted for the selenoester EDAG-1 at twice the concentration (1 µM), indicating the weaker activity of the EDAG-1 derivative. In addition, in the MCF-10A line, it was found that both tested selenoesters exhibited similar proapoptotic activity to MCF-7 cells, which highlights their significant toxicity to normal tissues. Apart from this, compound EDAG-8, especially at a concentration of 1 µM, causes an increase in the number of necrotic cells (Figure 5C), confirming its potent effect. Meanwhile, cisplatin induced the apoptotic process significantly weaker than the tested compounds in all cell lines. In addition, it was observed that while at a concentration of 0.5 µM, the percentage of the cell population with apoptosis induction was comparable between the two breast cancer cell lines for the compound EDAG-1, at a higher concentration (1 µM); this difference became pronounced with a more potent effect on MDA-MB-231 triple-negative breast cancer cells (Figure 5). Moreover, the selenoester EDAG-8 had a much more potent proapoptotic effect on MDA-MB-231 cells than MCF-7 cells even at half of the concentration—the apoptotic cell population was comparable for this compound at a concentration of 1 µM in MCF-7 cells and 0.5 µM in MDA-MB-231 cells. It is also noteworthy that triple-negative breast cancer cells (MDA-MB-231) were more sensitive to the tested compounds than MCF-7 cells, and the high proapoptotic activity of the selenoesters did not cause necrosis in the tested cancer cell lines (Figure 5).

Because caspase 8 is involved in the extrinsic pathway of apoptosis and affects the intrinsic pathway of this process [13,14,15], we examined the impact of novel selenoesters (EDAG-1 and EDAG-8) as well as cisplatin on caspase 8 activity in MCF-7 and MDA-MB-231 breast cancer cells following a 24 h exposure (concentrations of 0.5 and 1 µM) to those compounds. The findings of this research demonstrate that only the tested seleno-organic compounds significantly affect the activation of this protease in the tested cells (Figure 6). Depending on the cell line as well as the compound and its concentration, the population of cells with an active form of caspase 8 increased from about 1.8 to 6.1-fold as compared to the control. However, as before, EDAG-8 was the most active against both tested cell lines (especially against MDA-MB-231 cells), followed by EDAG-1. On the other hand, cisplatin did not cause a noticeable increase in the percentage of cells with an active form of caspase 8 (Figure 6). The overall conclusion emerging from this study is that the obtained data appropriately correlate with the results of the annexin V and PI double-staining assay (Figure 5), and the tested selenium compounds can induce apoptosis mediated by a DR-dependent pathway.

In the next step, we aimed to focus on the induction of apoptosis occurring via a mitochondrial-dependent (intrinsic) pathway; therefore, we evaluated the impact of the examined compounds (EDAG-1 and EDAG-8) and cisplatin on the mitochondrial membrane potential (∆Ψ_m_) and the proteins responsible for this process (Bax and Bcl-2) in breast cancer cells (MCF-7 and MDA-MB-231) following a 24 h exposure to the compounds (concentrations of 0.5 and 1 μM). The ∆Ψ_m_ assay results clearly show that the compounds EDAG-1 and EDAG-8 significantly decrease the mitochondrial membrane potential, especially in MDA-MB-231 triple-negative breast cancer cells (Figure 7). EDAG-8 was found to be the most active compound, while in the case of EDAG-1, the results were lower (Figure 7) but equally significant as compared to the control group. In turn, a statistically significant result for cisplatin was obtained only at a concentration of 1 µM in MDA-MB-231 cells. The other determinations for cisplatin oscillated more or less at the level of the control. In the case of the Bax protein study in the tested breast cancer cells, EDAG-8 was again the most active compound and its activity was notably evident in MDA-MB-231 triple-negative breast cancer cells (Figure 8). In contrast, a similar population of Bax positive cells compared to EDAG-8 at a concentration of 0.5 µM in MDA-MB-231 cells was observed for the selenoester EDAG-1 at twice the concentration (1 µM). In MCF-7 breast cancer cells, the selenoester EDAG-1 had a slightly weaker activity. In contrast to tested selenoesters, cisplatin did not significantly elevate the percentage of Bax positive cells at either concentration in either of the two tested cell lines (Figure 8).

Bcl-2 protein inhibits the tBid-dependent membrane-permeable activity of Bax. Thus, the activation and increase in Bax levels in cells should indicate a concurrent decrease in the anti-apoptotic peptide Bcl-2, which inhibits the decline of ΔΨ_m_ and apoptosis induction occurring via the intrinsic pathway [13]. These conclusions also translate into the results of our study to determine the level of Bcl-2 in MCF-7 and MDA-MB-231 breast cancer cells (Figure 9). The highest expression decrease in this protein under the impact of the tested selenium compounds was again observed in MDA-MB-231 triple-negative breast cancer. As in the previous study, cisplatin had no significant effect on the protein as compared to controls (Figure 9). The results of this study (∆Ψ_m_, Bax, and Bcl-2 protein expression—Figure 7, Figure 8 and Figure 9, respectively) correspond with those obtained earlier in the annexin V and PI double staining, which may indicate that the tested selenium compounds can induce an apoptotic process occurring through a mitochondrial-dependent (intrinsic) pathway.

With regard to the fact that caspase 9 is also one of the proteins of the intrinsic pathway of the apoptotic process [13,15], we analyzed its activity in breast cancer cells (MCF-7 and MDA-MB-231) under the impact of the tested seleno-organic compounds (EDAG-1 and EDAG-8) and cisplatin following a 24 h exposure to them (concentrations of 0.5 and 1 μM). The tested compounds caused a rise in the active form of this protease in both tested cell lines (Figure 10), except that in the case of the compound EDAG-1, the significant results were obtained only at a concentration of 1 µM compared to the control group. Meanwhile, the selenoester EDAG-8 had a more potent activity than the derivative EDAG-1 at half of the concentration. Similar values of active caspase 9 compared to EDAG-1 at a concentration of 1 µM were recorded for selenoester EDAG-8 at twice lower concentration (0.5 µM). Cisplatin did not significantly increase this protease activity. Both the obtained results and Figure 10 suggest that the observed increase in the active form of caspase 9 in MCF-7 (estrogen-dependent) and MDA-MB-231 (estrogen-independent) breast cancer cells confirms previous studies (AV/PI and changes of ΔΨ_m_ assays) and is consistent with their findings, allowing us to conclude that compounds EDAG-1 and EDAG-8 can trigger apoptosis occurring via an intrinsic pathway.

As is known, the effect of p53 on increasing the concentration of the proapoptotic protein Bax with concurrent inhibition of the Bcl-2 expression, which has anti-apoptotic properties, is prominent [14,16]. Since we had previously determined Bax and Bcl-2 protein levels, we also decided to evaluate p53 protein expression following a 24 h incubation with the selenoesters in this study (EDAG-1 and EDAG-8) and cisplatin, using concentrations of 0.5 and 1 µM, in breast cancer cells (MCF-7 and MDA-MB-231). A significant increase in p53 expression levels was noted only for the tested compounds (Figure 11). In both breast cancer cell lines, an upregulation of p53 with rising concentrations of the compounds under the study occurred. In contrast, cisplatin, as in previous assessments of other proteins and molecular processes, did not significantly affect p53 protein levels (Figure 11). As can be seen, the above results correspond well with previous studies and prove that the tested selenoorganic compounds induce elevation of p53 protein expression, which in turn enhances the levels of Bax peptide with a concurrent decrease in Bcl-2 concentration and subsequent mitochondrial membrane depolarization, as well as an increase in caspase 9 activity and the triggering of apoptosis occurring via the mitochondrial pathway.

To confirm previous studies, the last of the measurements performed involving the molecular mechanism of the apoptosis process was the assessment of the activity of caspase 3/7 in breast cancer cells (MCF-7 and MDA-MB-231) treated with the compounds under investigation (EDAG-1 and EDAG-8) and the conventionally used therapeutic agent (cisplatin) at 0.5 and 1 µM concentrations (treatment duration: 24 h). The highlighted point is that MCF-7 cells lack caspase 3 [17], so the results are lower than in the other tested cell line (MDA-MB-231) since only the active form of caspase 7 is determined. The obtained data show that the tested selenium compounds induced a significant elevation in caspase 3/7 activity mainly in MDA-MB-231 triple-negative breast cancer cells, while in MCF-7 cells, EDAG-8 only had an effect at a concentration of 1 µM (Figure 12). The compound with the highest activity against triple-negative breast cancer cells (MDA-MB-231), as before, was EDAG-8. Cisplatin, in the MCF-7 line, did not significantly increase caspase 7 activity. A similar situation was also noticed for the second tested cell line (MDA-MB-231). The analysis of previous studies allows us to see the correlation of the above results (Figure 12) with other data obtained from the analysis of apoptosis induction and the proteins involved in it, proving that the novel selenoesters EDAG-1 and EDAG-8 trigger the apoptosis process in breast cancer cells via two pathways (i.e., DR-dependent and mitochondrial pathways).

In addition to cytometric assays, we also carried out Western immunoblot analysis of the expression of Bax, caspase 7 and 9, and p53 in MCF-7 and MDA-MB-231 breast cancer cells treated (for 24 h) with the tested selenoesters (EDAG-1 and EDAG-8) and the reference agent (cisplatin)m which were applied at two concentrations (0.5 and 1 μM) to validate the results obtained by flow cytometry. The results obtained using this method validated previous findings (Figure 13; for original membrane images from the Western blot, see the Appendix A). As before, compound EDAG-8 proved to be the most active, especially against triple-negative breast cancer cells (MDA-MB-231). Meanwhile, the derivative EDAG-1 exhibited a slightly weaker effect. In contrast, in the case of cisplatin, the obtained bands were characterized by a similar intensity to the control (Figure 13).

### 2.3. EDAG-1 and EDAG-8 as Activators of Autophagy in Breast Cancer Cells

Autophagy is a process aimed at maintaining homeostasis in the cell. In the course of this process, redundant or non-functional molecules, cell fragments, or damaged organelles are degraded. However, excessive potentiation of this process, combined with the induction of apoptosis, may also become a strategy for enhancing the anticancer activity of a drug and/or overcoming the resistance of cancer cells to the therapy being applied [18]. Therefore, we have evaluated the impact of the tested compounds (EDAG-1 and ED-AG-8) and cisplatin on the induction of autophagy in estrogen-dependent and estrogen-independent breast cancer cells (MCF-7 and MDA-MB-231, respectively) and normal breast epithelial cells (MCF-10A) following a 24 h treatment with these substances at two concentrations—0.5 and 1 µM. Significant results have been reported only for the tested selenoesters (Figure 14), and EDAG-8, again, was the most active derivative. However, an approximately 2-fold weaker pro-autophagic activity compared to the most active derivative was characterized by the selenoester EDAG-1. Additionally, in MCF-10A cells, it was noticed that both selenoesters showed comparable induction of the autophagy process as in cancer cells. In the case of cisplatin, it did not exhibit the property of inducing autophagy in the tested cancer cells (Figure 14), and the obtained data were approximately at the level of the control group.

One of the most important targets of the PI3K/AKT/mTOR signaling pathway is the Akt protein, whose decline results in the downregulation of rapamycin (mTOR) kinase suppressive activity and the induction of autophagy [19,20]. Accordingly, to confirm the autophagy process occurring in the tested cells, we analyzed the level of Akt peptide in breast cancer cells (MCF-7 and MDA-MB-231) treated (treatment duration: 24 h) with the tested compounds (EDAG-1 and EDAG-8) and the conventionally used therapeutic agent (cisplatin) at 0.5 and 1 µM concentrations. A significant decline in Akt levels was recorded only in breast cancer cells treated with the tested selenoesters (Figure 15). Among all the tested compounds, as before, EDAG-8 most potently reduced Akt levels in the tested cell lines. Similar values for the population of Akt negative cells compared to EDAG-8 at a concentration of 0.5 µM were obtained for the selenoester EDAG-1 at the twice higher concentration (1 µM). In the previous study, cisplatin did not exhibit any pro-autophagic activity (Figure 14); therefore, there was no noticeable impact on the decrease in Akt, and the obtained results slightly differ from the control groups (Figure 15). In summary, the obtained Akt determination results correlate well with the data from the autophagy induction assay, which may indicate that the tested compounds (EDAG-1 and EDAG-8) may be activators of this process.

Autophagy is regulated by many different cellular mechanisms, one of which has already been discussed above. However, the activity of mTOR kinase is not only dependent on the Akt protein but is also affected by AMP-activated kinase (AMPK), which has a suppressive effect on that peptide [21]. Hence, in addition to investigating the levels of Akt, to confirm the test of autophagy induction, we undertook the analysis of AMPK kinase expression upon exposure to the compounds in the study (EDAG-1 and EDAG-8) and cisplatin (treatment duration: 24 h; concentrations of 0.5 and 1 µM) in breast cancer cells (MCF-7 and MDA-MB-231).

The highest activity of the tested derivatives was noticed in triple-negative breast cancer cells (MDA-MB-231), while in MCF-7 cells, only selenoester EDAG-8 showed significant results (Figure 16). Cisplatin, in all tested cell lines, did not significantly affect AMPK expression, and the results were almost at the level of the control (Figure 16). Thereby, as it can be seen, the obtained results correspond with the data obtained in the autophagy induction assay, so it can be concluded that the tested seleno-organic compounds can affect the activation of this process not via by the PI3K/Akt signaling pathway but also via AMPK kinase.

Among all the available autophagy markers, the level of LC3A/B protein is the most representative since, as is well known, it is essential for the formation of autophagosomes, and its increase correlates with an increase in the number of autophagosomes [22]. For this reason, we performed cytometric determination of LC3A/B protein levels after 24 h of breast cancer cell (MCF-7 and MDA-MB-231) treatment with the compounds in the study (EDAG-1 and EDAG-8) and the conventionally used therapeutic agent (cisplatin) at 0.5 and 1 µM concentrations. The analysis results have indicated that only the tested compounds EDAG-1 and EDAG-8 significantly changed the expression of this protein, especially in triple-negative breast cancer cells (MDA-MB-231; Figure 17). For the reference compound, the remaining results oscillated at the level of the control groups. In summary, the above results are consistent with the data obtained in the autophagy induction assay (Figure 14) and indicate that the selenoesters EDAG-1 and EDAG-8 are activators of this process, and that their potent anticancer activity and subsequent efficacy, especially against chemo-resistant MDA-MB-231 cells, may derive from their potential to concurrently trigger two different ways for cells to die, namely autophagy and apoptosis.

As with the proteins participating in apoptosis, we decided to verify the results obtained by flow cytometric analysis for proteins involved in the induction of autophagy. Western immunoblot analysis of Akt and AMPK proteins expression in breast cancer cells (MCF-7 and MDA-MB-231) exposed (for 24 h) to the EDAG-1 and EDAG-8 derivatives and cisplatin (0.5 and 1 µM) also confirmed, as before, the results obtained using the cytometric method (Figure 18; for original membrane images from the Western blotting, see the Appendix A). With an increase in the concentration of the tested selenoesters (EDAG-8 was the most active), there was an elevation in AMPK protein expression along with a corresponding fall in Akt levels, especially in MDA-MB-231 cells. However, the reference compound did not significantly affect the concentration of these proteins, which was comparable to the control (Figure 18).

## 3. Discussion

The high incidence of breast cancer, its increasing chemoresistance, and the associated increased mortality are high-priority problems for many clinicians, making them a prime premise for many researchers around the world to work in this field. Overcoming these problems and creating novel, more effective anticancer drugs is one of the challenges of modern oncology. Clinical trials and multiple meta-analyses have established that seleno-organic compounds are effective in the prevention of many cancer types, including breast cancer. Their molecular chemopreventive action, according to the available literature, occurs in multiple ways, i.e., through oxidative control of the cell (maintenance of oxidative–antioxidant homeostasis), suppression of neoangiogenesis and proliferation, as well as through cell cycle arrest, increased expression of the p53 protein (the most crucial suppressor of tumor transformation), and the subsequent induction of autophagy in cancer-altered cells [6,7]. It would seem that agents comprising this element in their structural design should exhibit the same or similar activity, making them potential drug candidates with anticancer activity. However, the literature data on Se compounds are still limited, especially as regards selenoesters, and their exact mode of action has not been completely comprehended and described to date.

Previous studies, including our recent one, have proven that Se compounds exhibit potent cytotoxic properties at relatively low doses against many types of cancer cells [8,9,10,12,23]. In this research, we have utilized two different breast cancer cell lines, namely MCF-7 cells, characterized by a generally good response to conventionally used therapies, and MDA-MB-231 cells (triple-negative breast cancer cells), that are mainly resistant or rapidly developing resistance to treatment in most cases. Our findings show that the selenoesters we examined are highly toxic against both tested breast cancer cell types (IC_50_ < 5 µM; except for EDAG-7 in the MDA-MB-231 line), while the most active ones (EDAG-1, EDAG-8) exhibit potent cytotoxic activity even at nanomolar concentrations (IC_50_ = 0.41–0.71 µM). However, they are not characterized by selectivity, neither in terms of breast cancer cells or MCF-10A normal breast epithelial cells, in which the IC_50_ values are similar to those obtained in cancer lines. We have already reached similar conclusions in our previous work [12], in which we evaluated a compound with a single selenoester group. In addition to selenoesters, among Se compounds, sodium selenite also has similar toxicity against normal bronchial epithelial cells (BEAS-2B) and lung cancer cells (A459) [24], and the toxicity of diphenyl diselenide depends on its dose and route of administration [8,23]. Problems of selectivity, toxicity to normal tissues, or resistance of tumor cells to drugs with anticancer activity may be solved by appropriate formulation of the drug and how it is administered. An example of such an approach is Doxil^®^, approved by the FDA for conventional use in 1995 against many types of cancer. A characteristic effect of anthracyclines, including doxorubicin, is cardiotoxicity. Encapsulating the drug molecules in polyethylene glycol (PEG)-coated liposomes significantly reduces doxorubicin’s toxicity to myocardial cells while increasing its anticancer efficacy [25].

Se and its compounds can “kill” cancer cells according to numerous diverse mechanisms. Among them, we distinguish apoptosis, the induction of which occurs even in small doses of various Se compounds [8,26]. This is confirmed by the findings of Nie et al. (2023) [27] and Chan et al. (2023) [28], in which various substances containing Se in their structure were used for triggering this type of cell death. Furthermore, in our previous study [12], we arrived at the same conclusions where the compound EDA-71 triggered the apoptotic process in over 50% of cells at a concentration of only 3 µM. This study conducted by our team also seems to confirm that. In addition, the triggering of this process proceeded in a dose-dependent manner, as at a concentration of 500 nM of the compounds EDAG-1 and EDAG-8, the value of apoptotic cells oscillated around 50% of the tested population, while increasing the concentration to 1 µM induced apoptosis in almost 80% of cells. Furthermore, in the case of MCF-10A normal human breast epithelial cells, the tested derivatives exhibited comparably high proapoptotic activity as in cancer cells. This may arise from the fact that the tested selenoesters caused very strong cell dysfunctionality, so there was an abrupt activation of proteins responsible for the process of apoptosis and its strong induction. Significantly, the compounds EDAG-1 and EDAG-8 (the most active compounds selected for further study after the viability test) did not affect the development of necrosis in the breast cancer cell lines used in this investigation. Therefore, the tested selenoesters may be good candidates for anticancer drugs in the future or provide a suitable framework for developing drugs with similar activity. The potent proapoptotic properties of the active substances account for an extremely desirable activity for future anticancer compounds, and the high cytotoxicity of the derivatives against normal tissues can be limited by using an appropriate formulation of the administered drug form, as discussed above.

In the extrinsic apoptotic pathway, mediated through DRs (death receptors), following the DISC complex formation, autocatalytic activation of caspase 8 occurs [14,15]. In our work, we have shown that selenoesters lead to upregulation of active caspase 8 in breast cancer cells (MCF-7 and MDA-MB-231), which has also been revealed by our previous study [12]. Moreover, our observations concerning the activity of this protease are consistent with the prior reports on the evaluation of various selenium compounds, suggesting that seleno-organic derivatives possess properties which stimulate the activity of this enzyme. Among the compounds upregulating caspase 8, we can distinguish, for example, selenadiazoles (HUVEC human umbilical vein endothelial cells and MCF-7 breast cancer cells), Se-methyl-*L*-selenocysteine (colon carcinoma cells, Colo205) [29], methylseleninic acid and sodium selenite (body-cavity-based lymphoma cells, BCBL1) [30], or ebselen (glioblastoma cells A172, T98G, and U87MG) [31]. Accordingly, it can be concluded that selenoesters EDAG-1 and EDAG-1 induce apoptosis through an extrinsic pathway associated with the activation of initiator caspase 8.

There are reports in the related literature that in addition to triggering apoptosis via the extrinsic pathway, many Se compounds affect the induction of this type of cell death mediated by the mitochondria (an intrinsic pathway) [8,12,23]. Mitochondrial membrane permeability is affected by both caspase 8 and the suppressor protein p53, which regulate the levels of Bax, a cell death-promoting protein, and Bcl-2, a protein that prevents the death of the cell. An increase in the expression of p53 results in a decrease in Bcl-2 concentration with a concurrent increase in Bax levels. Additionally, an increase in the activity of caspase 8 is followed by the tBid protein formation, which, along with increasing amounts of Bak and Bax in the mitochondrial membrane, oligomerize and generate macropores in it, leading to its destabilization and decrease in potential. The release of cytochrome c into the cell cytoplasm occurs due to this molecular mechanism. This leads to the formation of the apoptosome, where caspase 9 undergoes autocatalytic activation, in combination with procaspase 9, ATP, and Apaf-1. The occurrence of these events is succeeded by the beginning of the implementation stage of the apoptotic process within the cell, which involves the activation of execution caspases 3, 6, and 7 as a result of stimulation by caspases 8 and 9 [13,14,15]. Significantly, a drop in the cell’s mitochondrial membrane potential (ΔΨ_m_) occurs during the initial stage of the apoptotic process and is recognized as a marker of this event [32]. Soukupová and Rudolf, in their paper (2019) [33], report that treatment of RT112/D21-resistant bladder cancer cells with sodium selenite results in ΔΨ_m_ decline. Furthermore, treatment with the same compound in the study conducted by Qiao et al. [34] involving oral cancer xenografts in adult Wistar rats resulted in a reduced expression of Bcl-2 protein with a corresponding elevation in the Bax levels and caspase 9 activity. Similar observations have been reported for methylseleninic acid applied to cisplatin-resistant A459 lung cancer cells [35]. Additionally, this compound affects the upregulation of the p53 protein [36]. In turn, the action toward the aforementioned molecular targets of both sodium selenite and methylseleninic acid or other selenium compounds induces an increase in caspase 3 and 7 activity [8,27,30,33,34,35]. In our recent research, the findings obtained for the tested selenoesters EDAG-1 and EDAG-8 indicate that the Bax and p53 expression in the cell lines used in this study increases while the level of Bcl-2 protein concurrently declines, which occurs in a manner that depends on the dosage. In consequence, ΔΨ_m_ in breast cancer cells (MCF-7 and MDA-MB-231) is significantly reduced, especially when the compounds are used at a dose of 1 µM. This is important in the case of the BRAF^V600E^ mutation, which, although quite rare in triple-negative breast cancer (2–3% of cases), is characterized by poor patient survival prognosis [37]. This is associated, among other things, with the impact of this mutation on anti-apoptotic proteins existing on mitochondria and the correlated inhibition of mitochondrial membrane permeability (anti-apoptotic effect), causing resistance to the applied treatment [38]. Our tested compounds, even at a lower dose (0.5 µM), caused a sharp decline in ΔΨ_m_ (especially in MDA-MB-231 cells), related to an elevation in mitochondrial membrane permeability. This could suggest a potential reversal of the insensitivity to proapoptotic signaling induced by the BRAF^V600E^ mutation and the possible efficacy of therapy with these derivatives. Meanwhile, the further progression of the ΔΨ_m_ decrease intensifies the caspase 9 activation that, in turn, stimulates caspases 3 and 7. The marked rise in the caspase 9 active form is concurrent with a well-correlated elevation in the levels of executive caspases 3 and 7, which is definitive evidence that EDAG-1 and EDAG-8 derivatives exhibit their proapoptotic activity mediated through the mitochondrial pathway.

As previously mentioned, Se compounds can trigger various mechanisms leading to cell death. Autophagy is one of them, except for during apoptosis, as discussed above [8,12]. This process, also known as programmed cell death II, may be an additional component of the anticancer activity of compounds or have an additive effect on overcoming MDR in cancer cells. The main pathway regulating this process is PI3K/Akt/mTOR [19,20,39]. Xin and his team (2022) [35] noted the Akt protein inhibition in A549 lung cancer cells under the influence of methylseleninic acid, while the same observation was made by Woo et al. [40] in the case of their study of the effect of sodium selenite on JTIM-1-resistant breast cancer cells. Apart from this, AMPK protein can also regulate the level of mTOR kinase in the cell, thus the magnitude of the ongoing autophagy process [21]. The available related literature reports that Se compounds affect the AMPK/mTOR pathway. In the study evaluating Se-allylselenocysteine activity against HT-29 colon cancer cells, the authors observed that a decline in the expression of mTOR was related to an increase in the level of phosphorylated AMPK in the tested cells [41]. In addition, in all three studies [35,40,41], there was a marked elevation in the level of LC3 I/II, indicating enhanced formation of autophagosomes in cancer cells, which proved the activation of the autophagy process. The decrease in Akt levels and the increase in AMPK and LC3A/B protein expression in our study indicate the same thing, and confirm that Se compounds, including our studied selenoesters EDAG-1 and EDAG-8, can affect the PI3K/Akt/mTOR and AMPK/mTOR signaling pathways to induce autophagy in the tested breast cancer cells. We also reached similar conclusions in our previous article [12] investigating the effects of an EDA-71 mono-selenoester, where we found mTOR kinase inhibition. To summarize our studies on autophagy, the conclusion is that the compounds EDAG-1 and EDAG-8 may be activators of this process, which is one of the components of their potent anticancer activity.

The seleno-organic compounds belonging to the selenoesters group (EDA-71, EDAG-1, EDAG-8), and others with a nitrile end group (e.g., E-NS-4, EDAG-10), which we have investigated, exhibit interesting anticancer activity, and their effectiveness in fighting cancer is truly exceptional. The main conclusion from the analysis of our two previous works is that derivatives containing a ketone end group (EDA-71, EDAG-1, EDAG-8) are more active than those with a nitrile end fragment (E-NS-4, EDAG-7, EDAG-10). In addition, increasing the number of Se atoms in the structure of the selenoesters enhances its anticancer potential directed against breast cancer cells (MCF-7 and MDA-MB-231). Comparing the potency of apoptosis and autophagy induction in the tested breast cancer cell lines between mono-, di-, and tri-selenoesters, the conclusion is drawn that despite the use of a higher dose of mono-selenoester (EDA-71; 1.5 µM) efficacy, especially for the compound’s pro-autophagic properties, is lower than that of di-selenoester (EDAG-1) and much weaker than that of tri-selenoester (EDAG-8) at three times lower concentration (0.5 µM), particularly in MDA-MB-231 cells. In the context of apoptosis, the difference is small in the MCF-7 cell line, but significant, especially at lower concentrations (EDA-7 1.5 µM; EDAG-1/EDAG-8 0.5 µM), in triple-negative breast cancer cells (MDA-MB-231) [12]. Thus, the proapoptotic and pro-autophagic activity of the tested selenoesters with a ketone end fragment is as follows: mono-selenoesters (EDA-71) < di-selenoesters (EDAG-1) < tri-selenoesters (EDAG-8).

Figure 19 summarizes the potential molecular mechanism of action of the derivatives investigated in our studies conducted within the scope of this publication. However, these compounds’ detailed mode of action has not yet been completely comprehended, which guides new directions for further research. Unfortunately, the potent cytotoxicity found against normal cells limits their potential medical application in future oncological therapy, so as a first step, attention should be paid to its reduction. This could be achieved by appropriate drug formulation using targeted nanostructures while maintaining their potent anticancer activity. Further stages of the work, in turn, should include the determination of the specific molecular target of the tested derivatives, the way they enter the cells, the evaluation of the obtained anticancer drug formulation in vitro, and, with satisfactory results, the implementation of the therapy in an in vivo model. In addition to the above, it would be necessary to focus on conducting further thorough research on the molecular mode of anticancer action and to evaluate the activity of these compounds against other types of cancer. Moreover, the greater effectiveness of this treatment against resistant triple-negative breast cancer cells (MDA-MB-231) may indicate the potential MDR overcoming by our tested compounds, so performing studies in this area would be necessary.

## 4. Materials and Methods

### 4.1. Materials

Cisplatin, 3-(4,5-dimethylthiazol-2-yl)-2,5-diphenyltetrazolium bromide (MTT), dimethyl sulfoxide (DMSO), formaldehyde, glycine, methanol, sodium hydroxide, sodium dodecyl sulfate (SDS), Tris, HRP-labeled secondary anti-mouse antibody, and HRP-labeled secondary anti-rabbit antibody were purchased from Sigma-Aldrich (St. Louis, MO, USA). Ethanol and sodium chloride were purchased from Avantor Performance Materials (Poland), while hydrochloric acid and trichloroacetic acid (TCA) were provided by Chempur (Poland). 1,4-dithiothreitol (DTT) was obtained from Roche (Basel, Switzerland). Nitrocellulose membranes were obtained from BioRad Laboratories (Hercules, CA, USA). Amersham ECL Detection Reagents were from Cytiva (Marlborough, Massachusetts, USA). Stock cultures of human breast cancer cells (MCF-7 and MDA-MB-231) and normal human breast epithelial cells (MCF-10A) were received from the American Type Culture Collection (ATCC, Manassas, VA, USA). Dulbecco’s modified Eagle’s medium (DMEM), fetal bovine serum (FBS), the phosphate-buffered saline (PBS) used in the cell cultures, trypsin, glutamine, penicillin, and streptomycin were from Gibco (San Diego, CA, USA). An MEGM Mammary Epithelial Cell Growth Medium BulletKit was obtained from Lonza Bioscience (Basel, Switzerland). [^3^H]-thymidine (7 Ci/mmol) was provided from Moravek Biochemicals (Brea, CA, USA), and the Scintillation Cocktail Ultima Gold XR was from PerkinElmer (Waltham, MA, USA). The FITC Annexin V Apoptosis Detection Kit II, JC-1 MitoScreen Kit, and stain buffer were purchased from BD Pharmigen (San Diego, CA, USA). An Autophagy Assay kit, FAM-FLICA^®^ Caspase-3/7 Assay kit, FAM-FLICA^®^ Caspase-8 Assay kit, and FAM-FLICA^®^ Caspase-9 Assay kit were provided by ImmunoChemistry Technologies (Bloomington, MN, USA). Non-fat dried milk, β-actin mouse mAb, and p53 mouse mAb were obtained from Santa Cruz Biotechnology, Inc. (Dallas, TX, USA). AMPKβ mouse mAb and Akt mouse mAb were purchased from BD Transduction Laboratories (San Jose, CA, USA). Akt rabbit mAb (PE Conjugate), AMPKβ1/2 rabbit mAb, Bax rabbit mAb, Bax rabbit mAb (FITC Conjugate), Bcl-2 mouse mAb (Alexa Fluor^®^ 647 Conjugate), caspase-7 rabbit mAb, caspase-9 mouse mAb, LC3A/B XP^®^ Rrbbit mAb (Alexa Fluor^®^ 488 Conjugate), p53 rabbit mAb (Alexa Fluor^®^ 488 Conjugate), and Alexa Fluor^®^ 488-labeled secondary anti-rabbit antibody were obtained from Cell Signaling Technology (Beverly, MA, USA).

### 4.2. Tested Compounds

Four symmetrical Se compounds have been evaluated herein (Figure 2). These compounds are EDAG-1 [*Se*,*Se*-bis(2-oxopropyl) benzene-1,4-bis(carboselenoate)], EDAG-7 [*Se*,*Se*-bis(cyanomethyl) benzene-1,4-bis(carboselenoate)], EDAG-8 [*Se*,*Se*,*Se*-tris-(2-oxopropyl) benzene-1,3,5-tris-(carboxyselenoate)], and EDAG-10 [*Se*,*Se*,*Se*-tris-(cyanomethyl) benzene-1,3,5-tris-(carboxyselenoate)]. The four compounds have two or three selenoester moieties (-COSe-), bound to the aromatic benzene core of the molecule by the carbonyl of the selenoester. On the other hand, a functionalized alkyl moiety is bound to the selenium atom of the selenoester. The functional group of these alkyl branches of the molecule can be a ketone in the case of the compounds EDAG-1 and EDAG-8, or a nitrile, in the compounds EDAG-7 and EDAG-10. Finally, two compounds (EDAG-1 and EDAG-7) are symmetrical disubstituted derivatives with a 14-substitution pattern in the core benzene ring. The other two compounds (EDAG-8 and EDAG-10) are symmetrical trisubstituted selenoesters with a 1,3,5-substitution pattern in the benzene ring.

The synthesis and characterization of these four selenium-containing compounds have been reported in a patent application [42]. In brief, the procedure consists of 3 consecutive one-pot reactions performed in aqueous media. In the first one, selenium is reduced by a slow addition of sodium borohydride to generate sodium hydrogen selenide that is immediately used then to selenate the appropriate acyl chloride (in this case terephthaloyl chloride—compounds EDAG-1 and EDAG-7—or 1,3,5-benzenetricarbonyl chloride—compounds EDAG-8 and EDAG-10), forming an acylselenide intermediate. At this point, the reaction medium is quickly filtered to remove solid boron salts, and the filtrate with the intermediate acts as a nucleophile when adequate alkyl chloride is added (herein chloroacetone for compounds EDAG-1 and EDAG-8, or chloroacetonitrile to prepare the derivatives EDAG-7 and EDAG-10) to render the final compound. The desired derivative is filtered, washed, purified, and dried, as described [42]. The synthesis of close mono-selenated derivatives has been reported in previous works [9,43].

### 4.3. In Vitro Cell Culture of Normal and Cancerous Breast Cells

MCF-7 and MDA-MB-231 human breast cancer cell lines, as well as MCF-10A normal human breast epithelial cells, were acquired from the American Type Culture Collection (ATCC, Manassas, VA, USA). The characteristics of the breast cancer cell lines used in this study are compared in Table 1. MCF-7 and MDA-MB-231 cells were cultured in Dulbecco’s modified Eagle’s medium (Gibco, San Diego, CA, USA), while MCF-10A cells were cultured in the mammary epithelial cell growth medium with the following supplements: BPE, hEGF, insulin, hydrocortisone, GA-1000 (Lonza, Basel, Switzerland). All media were complemented by 10% of fetal bovine serum (FBS) and 1% of the following antibiotics: penicillin and streptomycin (both Gibco, San Diego, CA, USA). The cells were maintained in an incubator that provides the optimal growth conditions for the cell culture: 5% CO_2_, 37 °C, and humidity in a range of 90–95%. The cells were cultured in 100 mm plates (Sarstedt, Newton, NC, USA). Subsequently, after obtaining a subconfluent cell culture, the cells were detached with 0.05% trypsin with 0.02% EDTA (Gibco, San Diego, CA, USA). Then, a Scepter 3.0 handheld automated cell counter (Millipore, Burlington, MA, USA) was used for quantifying the number of cells that were subsequently seeded at a density of 5 × 10^5^ cells per well in six-well plates (“Nunc”) in 2 mL of the growth medium (Dulbecco’s modified Eagle’s medium or mammary epithelial cell growth medium, respectively). In this study, cells that obtained 80% confluence were used.

### 4.4. Cytotoxicity Test

The MTT assay was used to determine the cytotoxic effects of the compounds in the study on MCF-7 and MDA-MB-231 breast cancer cells as well as MCF-10A normal human breast epithelial cells based on Carmichael’s method. All cultured cell lines were treated with varied concentrations of the tested compounds in the medium (0.5–5 µM) and incubated for 24 h in an incubator providing optimal conditions for cell culture growth (37 °C temperature, 5% CO_2_, and 90–95% humidity) in 6-well plates (baseline seeding density: 5 × 10^5^ cells/well). After this time, the medium was removed, and the cell monolayer was washed with warm phosphate-buffered saline (PBS) without calcium and magnesium. In the next step, PBS and MTT solution in PBS (5 mg/mL) were added to each well to obtain a final MTT concentration of 0.5 mg/mL, and then the cells were incubated for 4 h (37 °C temperature, 5% CO_2_, 90–95% humidity). After the incubation period, the contents of each well were removed, 1 mL of DMSO was added to dissolve the formazan, and then the plate with the added DMSO was mixed for 5–10 min on a microplate shaker (Boeco, Hamburg, Germany). Then, 10 µL of Sorensen’s glycine buffer (a solution of 0.1 M glycine and 0.1 M sodium chloride adjusted to pH 10.5 with 0.1 M sodium hydroxide) was added. The absorbance of the obtained solution was immediately measured at a wavelength of λ = 570 nm using a Thermo Scientific Evolution 201 UV–VIS spectrophotometer (Thermo Fisher Scientific, Waltham, MA, USA). The absorbance result obtained in the control cells (without the addition of the tested compounds) was taken as 100%, while the viability of cells incubated with the tested compounds was presented as a percentage of the control value [12].

### 4.5. DNA Biosynthesis Assay

The study of the antiproliferative properties (intensity of [^3^H]-thymidine incorporation into cell DNA) of the tested derivatives against tested cell lines (MCF-7, MDA-MB-231, MCF-10A) was performed after 24 h incubation with varied concentrations of the tested compounds in the medium (0.5–5 µM) in 6-well plates (baseline seeding density: 5 × 10^5^ cells/well). After 24 h the medium was removed, the cells were washed with PBS (Corning, Kennebunk, ME, USA), and a fresh medium was added. After that, the cells were treated with 0.5 µCi of tritium-labeled thymidine (specific activity 7 Ci/mmol) by incubating them with this compound for 4 h under the same conditions as the cell culture. In the first step, the medium was removed, and the cell monolayer was washed twice with 1 mL of 0.05 M Tris-HCl buffer pH 7.4 containing 0.11 M NaCl. To denature proteins, the cells were washed twice with 1 mL of 5% trichloroacetic acid (TCA) solution. Finally, cell lysis was performed by adding 1 mL of 0.1 M sodium hydroxide (NaOH) solution containing 1% sodium dodecyl sulfate (SDS) to each well. After five minutes, the obtained cell lysates were transferred to scintillation vials with 2 mL of scintillation fluid added to them beforehand. Radioactivity was measured using the Scintillation Counter 1900 TR, TRI-CARB (Packard, Perkin Elmer, Inc., San Jose, CA, USA). The intensity of DNA biosynthesis in the analyzed cells was expressed in terms of dpm/well. The result of the radioactivity measurement in the control cells (without the addition of tested compounds) was taken as 100%, while the values of cells incubated with the tested compounds were presented as a percentage of the control value [12].

### 4.6. Evaluation of Apoptosis Induction by Flow Cytometry

MCF-7, MDA-MB-231, and MCF-10A cells were exposed (for 24 h) to EDAG-1 and DEAG-8 and a conventionally used anticancer agent (cisplatin) at concentrations 0.5 and 1 µM. The FITC Annexin V Apoptosis Detection Kit II (BD Pharmingen, San Diego, CA, USA) and a flow cytometer (BD FACSCanto II, BD Biosciences Systems, San Jose, CA, USA) were used in this evaluation. The assay was performed according to the manufacturer’s instructions. Flow cytometer calibration was performed by preparing two controls—a positive and a negative control. The positive control was cells in which apoptosis had been induced by using 2 µL of 3% formaldehyde in buffer and placing them on ice for 30 min. The negative control was cells that were not treated with any of the proapoptotic agents. First, in the cells treated with the tested compounds as well as the controls, the medium was removed, and the cells were washed twice with cold PBS. Subsequently, the cells were resuspended in the binding buffer included in the kit at a concentration of 1 × 10^6^ cells/mL. From each sample, 100 µL of cell suspension was taken and transferred to test tubes, to which 5 µL each of FITC annexin V and propidium iodide (PI) were then added. The contents of the test tubes were gently vortexed and incubated for 15 min at room temperature, protected from light. After the required time, the contents of the test tubes were made up to 500 µL with binding buffer and immediately analyzed in a flow cytometer (10,000 events measured). After the flow cytometer readout, the results were analyzed using the FACSDiva 6.0 software (BD Biosciences Systems, San Jose, CA, USA). The equipment was calibrated with the BD Cytometer Setup and Tracking Beads (BD Biosciences, San Diego, CA, USA).

### 4.7. Evaluation of Mitochondrial Membrane Polarity Changes (ΔΨ_m_) by Flow Cytometry

The effects of EDAG-1 and EDAG-8 as well as cisplatin (treatment duration: 24 h; concentrations: 0.5 and 1 µM) on mitochondrial membrane polarity changes in two breast cancer cell lines (MCF-7 and MDA-MB-231) were analyzed by flow cytometry (flow cytometer BD FACSCanto II, BD Biosciences Systems, San Jose, CA, USA) and using a JC-1 MitoScreen kit (BD Pharmigen, San Diego, CA, USA). The entire cell staining and cytometric analysis procedures were performed according to the instructions provided with the kit. In the first step, the cells (1 × 10^6^ cells/sample) were washed and resuspended in 0.5 mL of buffer containing 10 µg/mL JC-1 dye. Incubation was carried out for 15 min at room temperature, protected from light. Afterward, the cells were washed twice with buffer, resuspended in 0.5 mL PBS, and immediately analyzed using a flow cytometer (BD FACSCanto II; 10,000 events measured) and the FACSDiva 6.0 software (BD Biosciences Systems, San Jose, CA, USA) to count the percentage of cells with reduced ΔΨ_m_. The equipment was calibrated with the BD Cytometer Setup and Tracking Beads (BD Biosciences, San Diego, CA, USA).

### 4.8. Evaluation of Activity of Caspase 3/7, 8, and 9 by Flow Cytometry

The FAM-FLICA^®^ Caspase Assay kits (all from ImmunoChemistry Technologies, Bloomington, MN, USA) and flow cytometer (BD FACSCanto II, BD Biosciences Systems, San Jose, CA, USA) were used to evaluate the activity of caspase 3/7, 8, and 9. MCF-7 and MDA-MB-231 cells were treated with selenoesters under the study (EDAG-1 and EDAG-8) and a reference agent (cisplatin) at two different concentrations (0.5 and 1 µM; treatment duration: 24 h). Following this, cells were collected, washed twice with cold PBS, and resuspended in an Apoptosis Wash Buffer to a final concentration of 5 × 10^5^ cells/mL. In the next step, 290 µL each of cell suspension was taken and transferred into tubes. Then 10 µL of FLICA solution diluted immediately before use (1:5 *v*/*v*, using PBS) was added to the cells, mixed by pipetting, and incubated in the dark for 1 h at 37 °C. After this time, the cells were washed twice with 2 mL Apoptosis Wash Buffer, centrifuged, and resuspended in 300 µL of the buffer. Thus, prepared samples were immediately analyzed using a BD FACSCanto II flow cytometer (10,000 events) with the FACSDiva 6.0 software (both from BD Biosciences Systems, San Jose, CA, USA). The equipment calibration was carried out using the BD Cytometer Setup and Tracking Beads (BD Biosciences, San Diego, CA, USA).

### 4.9. Evaluation of Autophagy Induction by Flow Cytometry

The effects of the novel selenoesters and a reference compound (cisplatin) at concentrations of 0.5 and 1 µM (after 24 h) on the activation of the autophagy process in MCF-7, MDA-MB-231, and MCF-10A cells were evaluated utilizing the Autophagy Assay kit, Red (ImmunoChemistry Technologies, Bloomington, MN, USA) via flow cytometry (flow cytometer from BD FACSCanto II, BD Biosciences Systems, San Jose, CA, USA). The entire assay was performed according to the manufacturer’s instructions. After drug treatment, unfixed cells were washed and resuspended in PBS at a concentration of 5 × 10^5^ cells/mL. Then, 490 µL each of the cell suspension was taken, transferred to test tubes, and 10 µL of Autophagy Probe, Red solution (previously diluted 1:5 in PBS) was added and incubated (30 min, 37 °C, in the dark). After incubation, the cells were washed and resuspended in cellular assay buffer, before finally adding fixative at a ratio of 1:5 (*v*/*v*). After this step, the prepared samples were immediately measured using a flow cytometer (BD FACSCanto II; 10,000 events measured), and the percentage of cells with an occurring autophagy process was calculated using the FACSDiva 6.0 software (both from BD Biosciences Systems, San Jose, CA, USA). The equipment calibration was performed using the BD Cytometer Setup and Tracking Beads (BD Biosciences, San Diego, CA, USA).

### 4.10. Akt, Bax, Bcl-2, LC3A/B, and p53 Protein Determination by Flow Cytometry

The levels of Akt, Bax, Bcl-2, LC3A/B, and p53 proteins in MCF-7 and MDA-MB-231 cells were assessed using flow cytometry after treatment (24 h) with the compounds EDAG-1, EDAG-8, and cisplatin at concentrations of 0.5 and 1 µM. For this purpose, Akt (PE), Bax (FITC), Bcl-2 (Alexa Fluor^®^ 647), LC3A/B (Alexa Fluor^®^ 488), and p53 (Alexa Fluor^®^ 488) antibodies conjugated to different fluorochromes (all from Cell Signaling Technology, Beverly, MA, USA) were utilized. Each assay was performed according to the manufacturer’s protocol available on their website. After incubation with the tested compounds, the cells were centrifuged, resuspended in 4% formaldehyde (100 µL/1 × 10^6^ cells), and fixed for 15 min at room temperature. Afterward, to remove the formaldehyde, the cells were washed by centrifugation with excess PBS and the permeabilization step was triggered. For this purpose, ice-cold 90% methanol was added to the pre-chilled cell pellet, gently vortexed, and incubated for 1 h on ice. Later, the cells were washed by centrifugation with excess PBS again, resuspended in 100 µL of 1:100 diluted primary antibody (stain buffer was used for dilution), and incubated for 1 h at room temperature in the dark. After incubation, the cells were washed, resuspended in 300 µL PBS, and immediately tested with a flow cytometer (BD FACSCanto II, BD Biosciences Systems, San Jose, CA, USA; 10,000 events measured). The obtained results were analyzed using the FACSDiva 6.0 software (BD Biosciences Systems, San Jose, CA, USA). In the case of the p53 protein, the MRF values were obtained first during the flow cytometer readout utilizing FACSDiva software and later analyzed using FCS Express 7 (De Novo Software, Pasadena, CA, USA). The equipment calibration was performed using the BD Cytometer Setup and Tracking Beads (BD Biosciences, San Diego, CA, USA).

### 4.11. AMPK Protein Determination by Flow Cytometry

Identification of AMPK protein was carried out after incubation (24 h) of two breast cancer cell lines (MCF-7 and MDA-MB-231) with the compounds in the study (EDAG-1, EDAG-8, and cisplatin; concentrations of 0.5 and 1 µM). In this assay, a primary anti-AMPKβ1/2 antibody and a secondary anti-rabbit IgG Alexa Fluor^®^ 488 conjugate (both from Cell Signaling Technology, Beverly, MA, USA) were used. The assay was performed according to the manufacturer’s protocol available on their website. After incubation with the tested compounds, the cells were centrifuged, resuspended in 4% formaldehyde (100 µL/1 × 10^6^ cells), and fixed for 15 min at room temperature. Afterward, to remove the formaldehyde, the cells were washed by centrifugation with excess PBS and the permeabilization step was triggered. For this purpose, ice-cold 90% methanol was added to the pre-chilled cell pellet, gently vortexed, and incubated for 1 h on ice. Later, the cells were washed by centrifugation with excess PBS again, resuspended in 100 µL of 1:100 diluted primary antibody (stain buffer was used for dilution), and incubated for 1 h at room temperature in the dark. After the required incubation period, the cells were washed, resuspended in 100 µL of secondary antibody diluted in stain buffer (1:100 ratio), and incubated for 30 min at room temperature, protected from light. Subsequently, the cells were washed, resuspended in 300 µL PBS, and immediately tested with a flow cytometer (BD FACSCanto II, BD Biosciences Systems, San Jose, CA, USA; 10,000 events measured). The obtained results were analyzed using the FACSDiva 6.0 software (BD Biosciences Systems, San Jose, CA, USA) and then using FCS Express 7 (De Novo Software, Pasadena, CA, USA). The equipment calibration was performed using the BD Cytometer Setup and Tracking Beads (BD Biosciences, San Diego, CA, USA).

### 4.12. Western Immunoblotting

MCF-7 and MDA-MB-231 cells were exposed to EDAG-1, EDAG-8, and a clinically used agent (cisplatin) at two concentrations (0.5 and 1 µM) for 24 h, and then harvested using cell lysis buffer supplemented with a protease/phosphatase inhibitor cocktail. Protein concentrations in the samples were determined using the Lowry method [48]. Samples were prepared containing equal amounts of protein (30 µg/lane) and an appropriate amount of Laemmli buffer (120 mM Tris-HCl, 20% glycerol, 0.4% SDS, and 0.02% bromophenol blue, pH 6.8) containing 100 mM 1,4-dithiothreitol (DTT; Roche, Basel, Switzerland). Samples were denatured at 95 °C for 10 min. The proteins were separated using the SDS-PAGE method described by Laemmli [49]. Proteins were separated on 12% SDS-PAGE gels. After this step, the gels were washed in cold Towbin buffer (25 mM Tris, 192 mM glycine, 20% (*v*/*v*) methanol, 0.025–0.1% SDS, pH 8.3) and were transferred to 0.2-µm nitrocellulose membranes (BioRad Laboratories, Hercules, CA, USA) using the Trans-Blot apparatus (BioRad Laboratories, Hercules, CA, USA). Transfer conditions were 250 mA, at 4–8 °C, overnight in freshly prepared Towbin buffer. Membranes were blocked with 5% non-fat dried milk (Santa Cruz Biotechnology, Dallas, Texas, USA) in TBS-T (20 mM Tris, 150 mM NaCl, 0.1% Tween-20, pH 7.6) for 1 h at room temperature with gentle shaking. After the blocking step, membranes were washed four times with TBS-T (4 × 15 min) and incubated with primary monoclonal antibodies (against β-actin, p53 (both from Santa Cruz Biotechnology, Dallas, TX, USA), Akt, AMPK (both from BD Transduction Laboratories, San Jose, CA, USA), Bax, caspase 7, and caspase 9 (all from Cell Signaling Technology, Danvers, MA, USA)). Membranes were incubated with primary antibodies overnight at 4 °C. Then, the membranes were again washed four times with TBS-T (4 × 15 min) and incubated with the appropriate secondary antibodies conjugated with HRP (Sigma-Aldrich, Saint Louis, MI, USA) at a concentration of 1:1000 in 5% non-fat milk dried in TBS-T for 1 h at room temperature with gentle shaking. Membranes were washed four times with TBS-T (4 × 15 min) and visualized using Amersham ECL Detection Reagents (Cytiva, Marlborough, MA, USA). Images were taken using BioSpectrum Imaging System UVP (Ultra-Violet Products Ltd., Cambridge, UK), whereas densitometric analysis was performed using UVIBand software, version 15.08d (Uvitec Ltd., Cambridge, UK).

### 4.13. Statistical Analysis

The findings are shown as the mean ± standard deviation (SD) of a minimum of three (n = 3) independent investigations, each performed in triplicate. The statistical analysis of the received data was carried out using the GraphPad Prism 8 software (GraphPad Software, San Diego, CA, USA). To ascertain the statistical significance between the experimental groups (treated cells) and control group (untreated cells), Dunnett’s test followed by a one-way ANOVA was employed. A *p*-value of less than 0.05 was defined as statistically significant.

## 5. Conclusions

In our work, we investigated the effects of four novel seleno-organic compounds on human breast cancer cells (MCF-7 and MDA-MB-231). The obtained results indicated that the tested selenoesters (especially EDAG-1 and EDAG-8) exhibited high toxic and antiproliferative potential against the tested cell lines (even at the nanomolar range). Further molecular studies of the two most active compounds (EDAG-1 and EDAG-8) showed that their potent anticancer activity was associated with the activation of both autophagy (↓ Akt, ↑ autophagosomes and autolysosomes, AMPK, LC3A/B) as well as apoptosis (↑ apoptotic cells and caspase 3/7 activity) occurring via extrinsic (↑ caspase 8 activity) and intrinsic (↓ Bcl-2 expression and ΔΨm, ↑ levels of Bax and p53 protein, as well as caspase 9 activity) signaling pathways.

To summarize the above, the novel selenoesters (EDAG-1 and EDAG-8) exhibit uniquely interesting anticancer activity, especially against treatment-resistant triple-negative breast cancer cells. This suggests that these derivatives may be promising potential drug candidates for use in future breast cancer therapy; however, their exact molecular mode of action requires further research in this field.

## Figures and Tables

**Figure 1 ijms-25-07764-f001:**
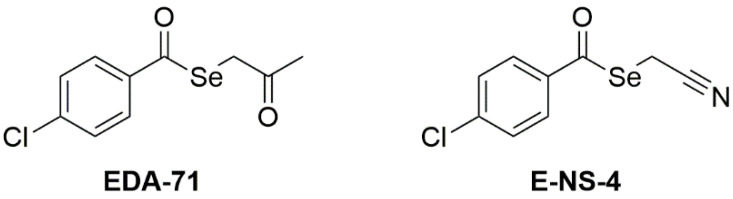
The chemical structures of the compounds that we have recently investigated [12].

**Figure 2 ijms-25-07764-f002:**
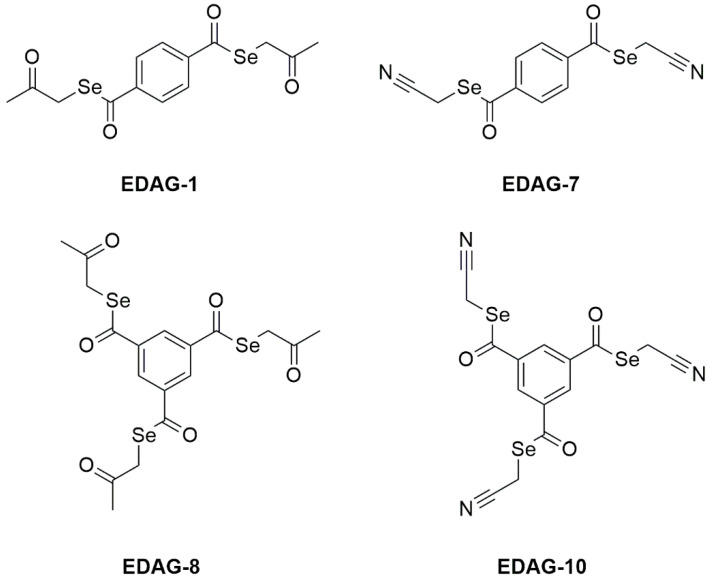
Chemical structure of the tested compounds.

**Figure 3 ijms-25-07764-f003:**
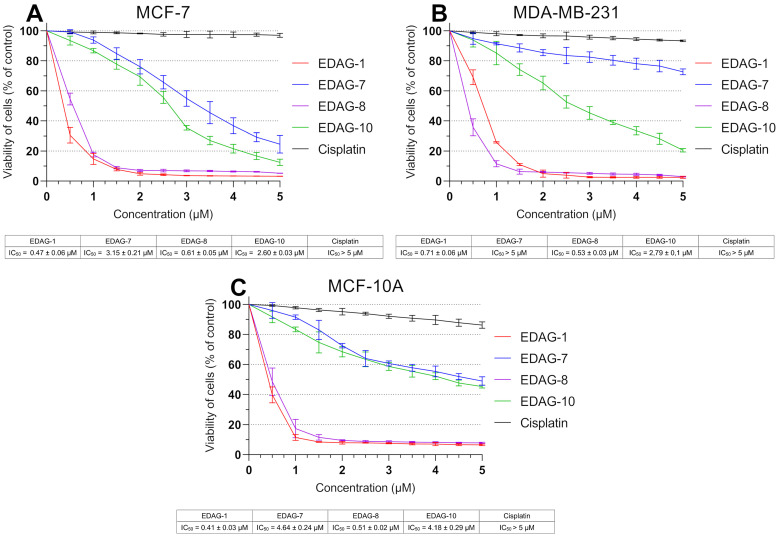
Viability of breast cancer cells ((**A**)—MCF-7; (**B**)—MDA-MB-231) and normal breast epithelial cells ((**C**)—MCF-10A) after 24 h incubation with the compounds under study and cisplatin at different concentrations. The mean values (expressed as the percentage of untreated cells) with standard deviation (SD) were reported based on data from three separate experiments (n = 3) carried out in triplicate.

**Figure 4 ijms-25-07764-f004:**
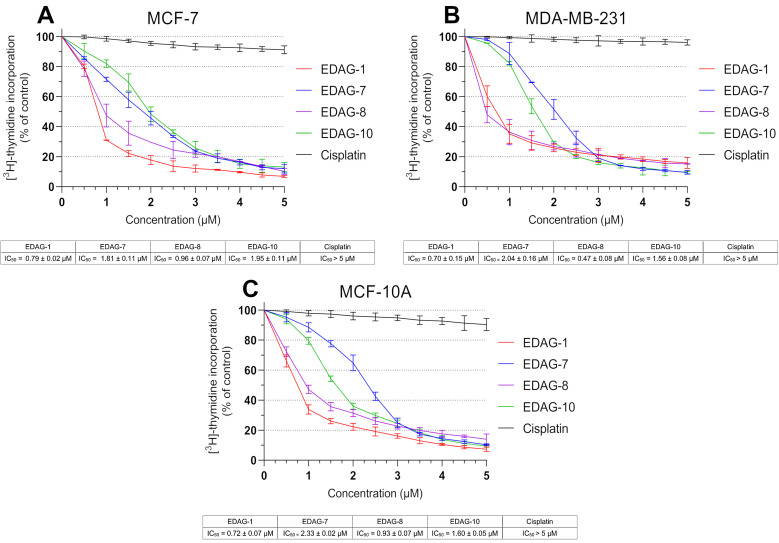
DNA biosynthesis process observed in breast cancer cells ((**A**)—MCF-7; (**B**)—MDA-MB-231) and normal breast epithelial cells ((**C**)—MCF-10A) after 24 h incubation with the compounds under study and cisplatin at different concentrations. The mean values (expressed as the percentage of untreated cells) with SD were reported based on data from three separate experiments (n = 3) carried out in triplicate.

**Figure 5 ijms-25-07764-f005:**
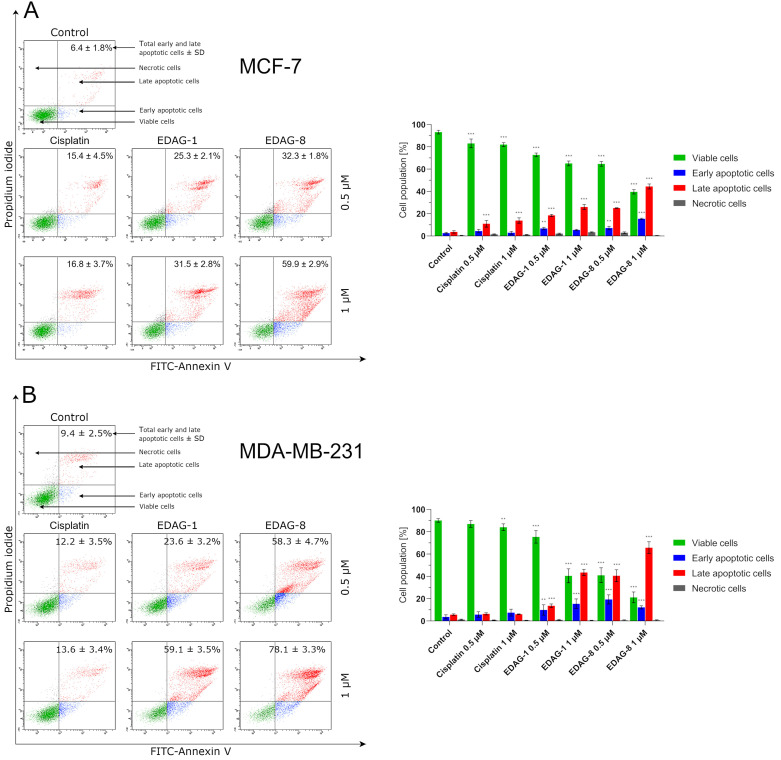
Apoptosis induction in breast cancer cells ((**A**)—MCF-7; (**B**)—MDA-MB-231) and normal breast epithelial cells ((**C**)—MCF-10A) following a 24 h exposure to the tested compounds (EDAG-1 and EDAG-8) and cisplatin (concentrations of 0.5 and 1 μM). Annexin V-FITC/propidium iodide double staining and a flow cytometer were used in the experiment. The mean values with SD were reported based on data from three separate experiments (n = 3) conducted in triplicate. Statistical differences between the experimental groups (treated cells) and control (untreated cells) were assessed using one-way ANOVA and Dunnett’s test. ** *p* < 0.01 vs. control group, *** *p* < 0.001 vs. control group.

**Figure 6 ijms-25-07764-f006:**
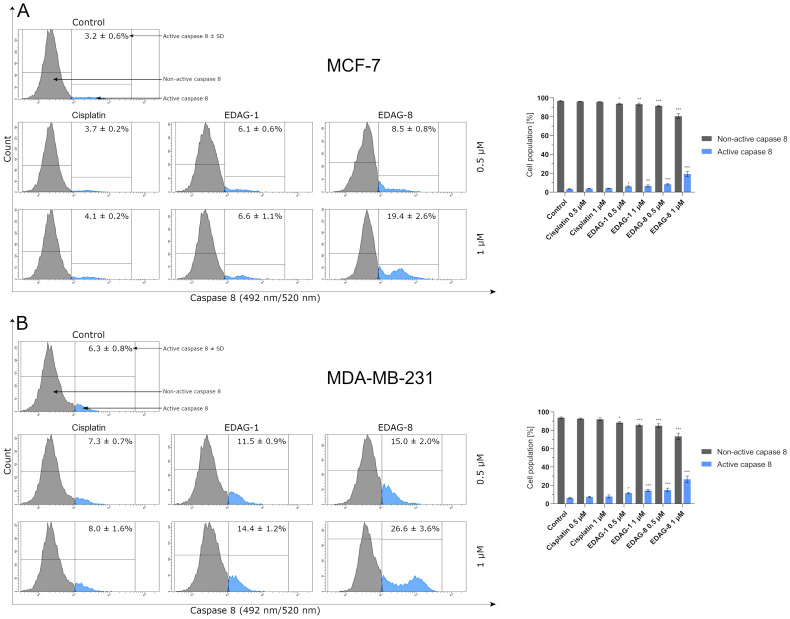
The activity of caspase 8 in breast cancer cells ((**A**)—MCF-7; (**B**)—MDA-MB-231) following a 24 h exposure to the tested compounds (EDAG-1 and EDAG-8) and cisplatin (concentrations of 0.5 and 1 μM). The FAM-FLICA^®^ Caspase-8 Assay Kit and a flow cytometer were used in the experiment. The mean values with SD were reported based on data from three separate experiments (n = 3) conducted in triplicate. Statistical differences between the experimental groups (treated cells) and control (untreated cells) were assessed using one-way ANOVA and Dunnett’s test. * *p* < 0.05 vs. control group, ** *p* < 0.01 vs. control group, *** *p* < 0.001 vs. control group.

**Figure 7 ijms-25-07764-f007:**
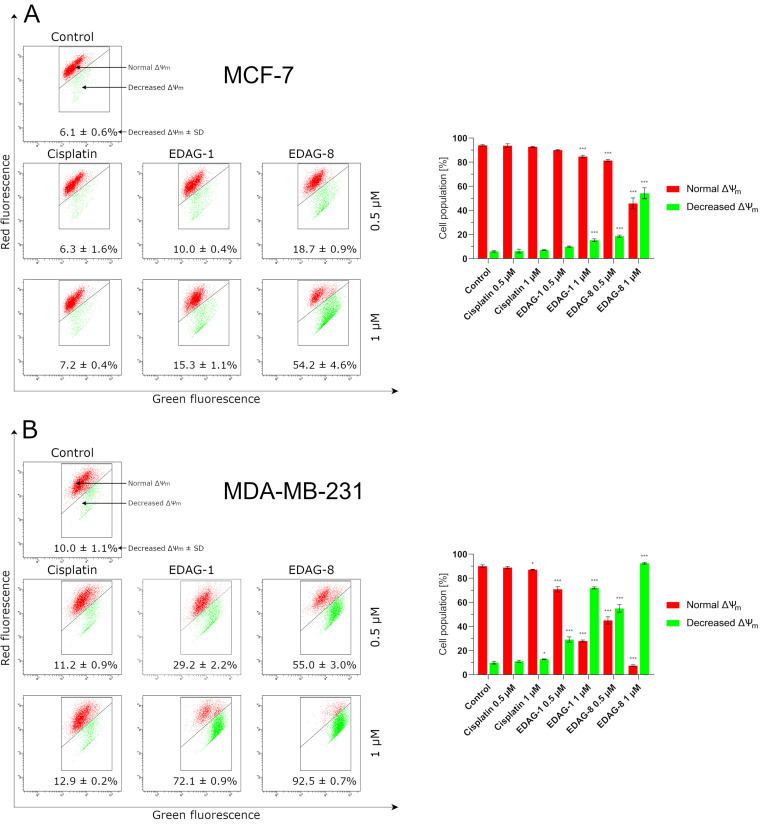
Mitochondrial membrane potential (∆Ψ_m_) changes in breast cancer cells ((**A**)—MCF-7; (**B**)—MDA-MB-231) following a 24 h exposure to the tested compounds (EDAG-1 and EDAG-8) and cisplatin (concentrations of 0.5 and 1 μM). The JC-1 staining and a flow cytometer were used in the experiment. The mean values with SD were reported based on data from three separate experiments (n = 3) conducted in triplicate. Statistical differences between the experimental groups (treated cells) and control (untreated cells) were assessed using one-way ANOVA and Dunnett’s test. * *p* < 0.05 vs. control group, *** *p* < 0.001 vs. control group.

**Figure 8 ijms-25-07764-f008:**
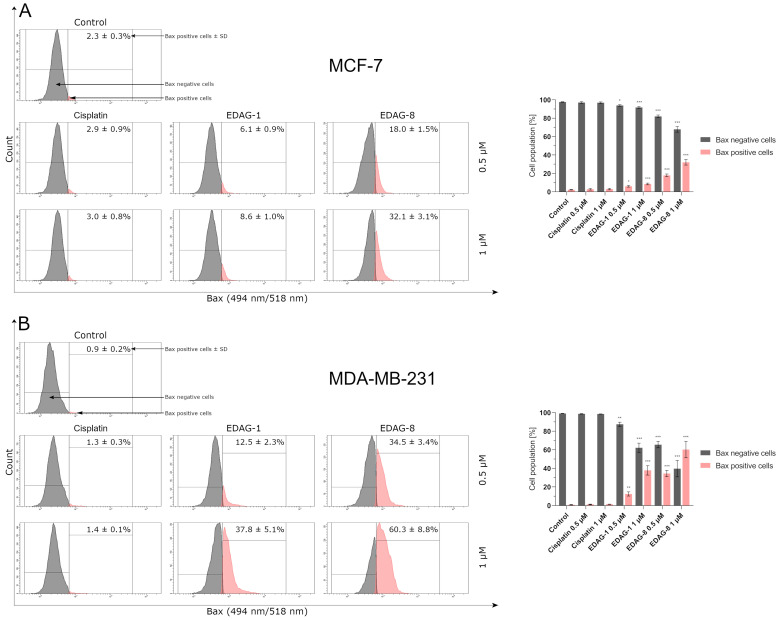
Bax protein levels in breast cancer cells ((**A**)—MCF-7; (**B**)—MDA-MB-231) following a 24 h exposure to the tested compounds (EDAG-1 and EDAG-8) and cisplatin (concentrations of 0.5 and 1 μM). The FITC conjugate with an anti-Bax antibody and a flow cytometer were used in the experiment. The mean values with SD were reported based on data from three separate experiments (n = 3) conducted in triplicate. Statistical differences between the experimental groups (treated cells) and control (untreated cells) were assessed using one-way ANOVA and Dunnett’s test. * *p* < 0.05 vs. control group, ** *p* < 0.01 vs. control group, *** *p* < 0.001 vs. control group.

**Figure 9 ijms-25-07764-f009:**
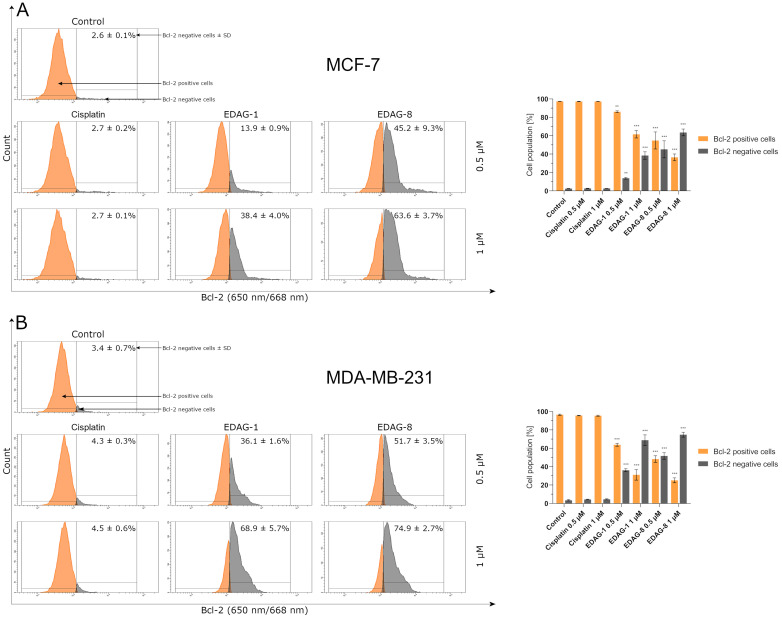
Bcl-2 protein levels in breast cancer cells ((**A**)—MCF-7; (**B**)—MDA-MB-231) following a 24 h exposure to the tested compounds (EDAG-1 and EDAG-8) and cisplatin (concentrations of 0.5 and 1 μM). Alexa Fluor^®^ 647 conjugate with an anti-Bcl-2 antibody and a flow cytometer were used in the experiment. The mean values with SD were reported based on data from three separate experiments (n = 3) conducted in triplicate. Statistical differences between the experimental groups (treated cells) and control (untreated cells) were assessed using one-way ANOVA and Dunnett’s test. ** *p* < 0.01 vs. control group, *** *p* < 0.001 vs. control group.

**Figure 10 ijms-25-07764-f010:**
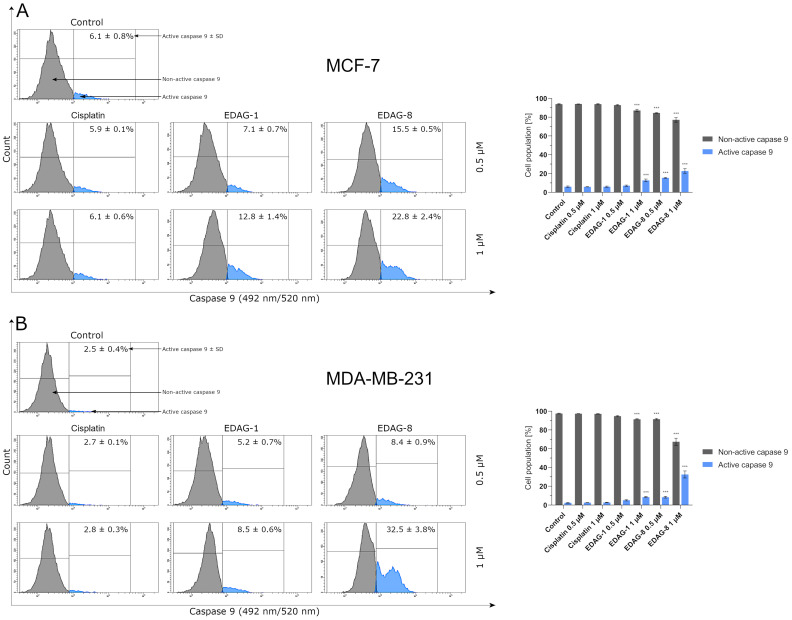
The activity of caspase 9 in breast cancer cells ((**A**)—MCF-7, (**B**)—MDA-MB-231) following a 24 h exposure to the tested compounds (EDAG-1 and EDAG-8) and cisplatin (concentrations of 0.5 and 1 μM). The FAM-FLICA^®^ Caspase-9 Assay kit and a flow cytometer were used in the experiment. The mean values with SD were reported based on data from three separate experiments (n = 3) conducted in triplicate. Statistical differences between the experimental groups (treated cells) and control (untreated cells) were assessed using one-way ANOVA and Dunnett’s test. *** *p* < 0.001 vs. control group.

**Figure 11 ijms-25-07764-f011:**
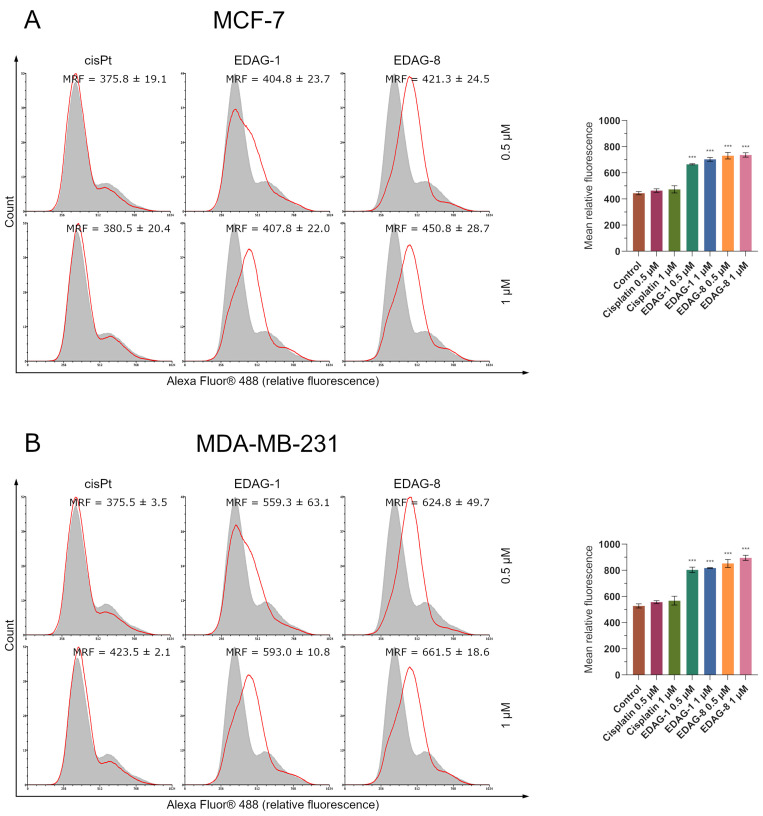
P53 protein expression in breast cancer cells ((**A**)—MCF-7; (**B**)—MDA-MB-231) following a 24 h exposure to the tested compounds (EDAG-1 and EDAG-8) and cisplatin (concentrations of 0.5 and 1 μM). The grey histogram represents the fluorescence of the control group, and the red colored one represents the experimental group. The fluorescence’s difference (value shift) shows the corresponding protein activity. Alexa Fluor^®^ 488 conjugate with an anti-p53 antibody and a flow cytometer were used in the experiment. The mean values with SD were reported based on data from three separate experiments (n = 3) conducted in triplicate. Statistical differences between the experimental groups (treated cells) and control (untreated cells) were assessed using one-way ANOVA and Dunnett’s test. *** *p* < 0.001 vs. control group.

**Figure 12 ijms-25-07764-f012:**
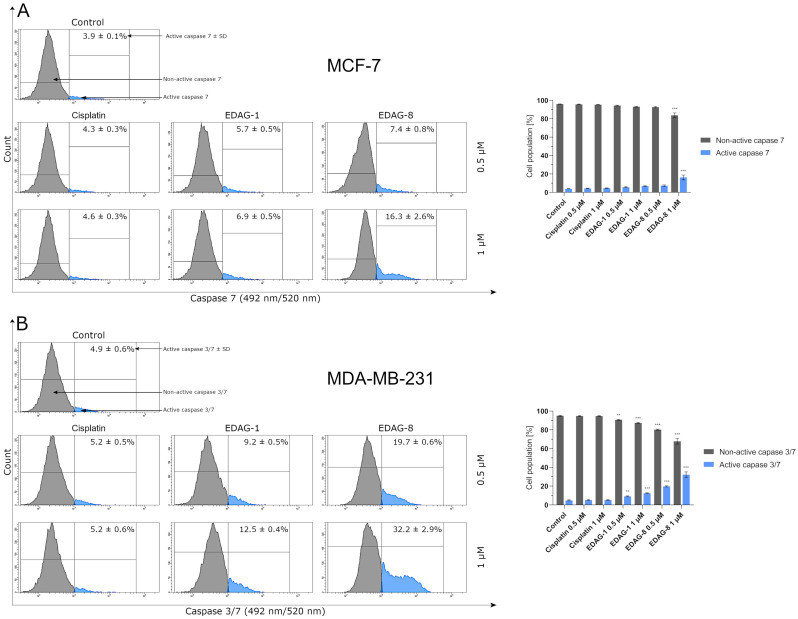
The activity of caspase 3/7 in breast cancer cells ((**A**)—MCF-7; (**B**)—MDA-MB-231) following a 24 h exposure to the tested compounds (EDAG-1 and EDAG-8) and cisplatin (concentrations of 0.5 and 1 μM). The FAM-FLICA^®^ Caspase-3/7 Assay Kit and a flow cytometer were used in the experiment. The mean values with SD were reported based on data from three separate experiments (n = 3) conducted in triplicate. Statistical differences between the experimental groups (treated cells) and control (untreated cells) were assessed using one-way ANOVA and Dunnett’s test. ** *p* < 0.01 vs. control group, *** *p* < 0.001 vs. control group.

**Figure 13 ijms-25-07764-f013:**
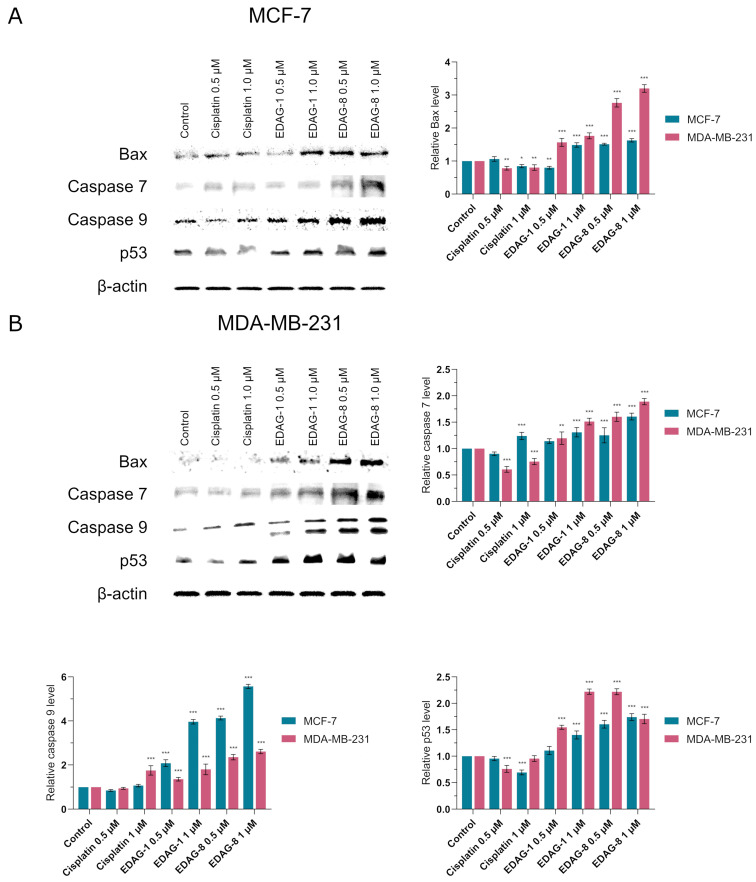
Western immunoblot analysis of the expression of Bax, caspase 7 and 9, as well as p53 in breast cancer cells ((**A**)—MCF-7; (**B**)—MDA-MB-231) following a 24 h exposure to the tested compounds (EDAG-1 and EDAG-8) and cisplatin (concentrations of 0.5 and 1 μM). For the performed experiment, equal amounts (30 µg/lane) of protein lysates were used. The intensity of band staining was quantified by densitometric analysis. The mean values with SD were reported based on data from three separate experiments (n = 3) conducted in triplicate. Statistical differences between the experimental groups (treated cells) and control (untreated cells) were assessed using one-way ANOVA and Dunnett’s test. * *p* < 0.05 vs. control group, ** *p* < 0.01 vs. control group, *** *p* < 0.001 vs. control group.

**Figure 14 ijms-25-07764-f014:**
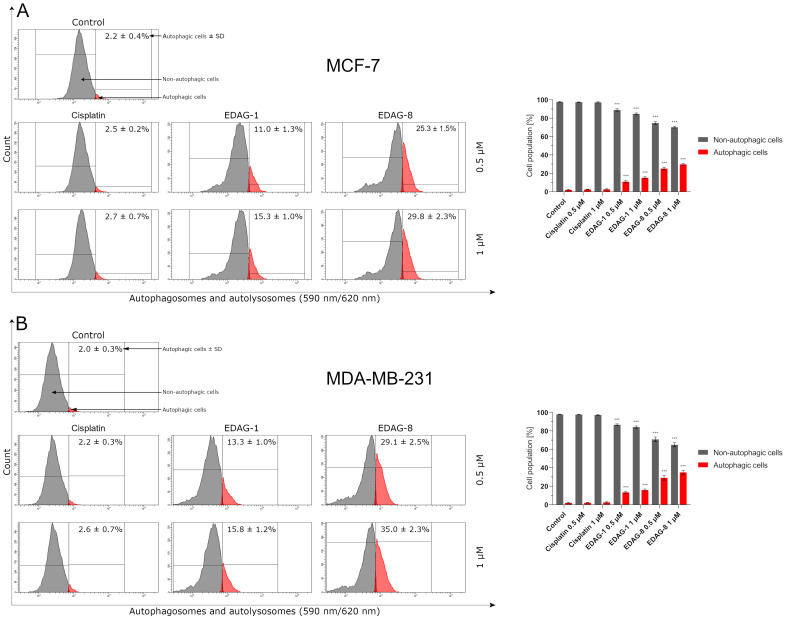
Autophagy induction in breast cancer cells ((**A**)–MCF-7; (**B**)–MDA-MB-231) and normal breast epithelial cells ((**C**)—MCF-10A) following a 24 h exposure to the tested compounds (EDAG-1 and EDAG-8) and cisplatin (concentrations of 0.5 and 1 μM). An Autophagy Assay Kit and a flow cytometer were used in the experiment. The mean values with SD were reported based on data from three separate experiments (n = 3) conducted in triplicate. Statistical differences between the experimental groups (treated cells) and control (untreated cells) were assessed using one-way ANOVA and Dunnett’s test. *** *p* < 0.001 vs. control group.

**Figure 15 ijms-25-07764-f015:**
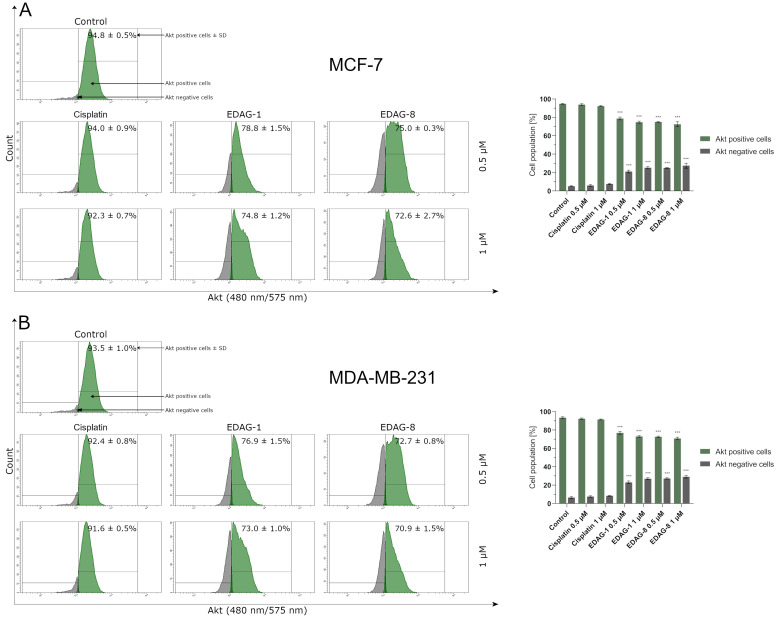
Akt protein levels breast cancer cells ((**A**)—MCF-7; (**B**)—MDA-MB-231) following a 24 h exposure to the tested compounds (EDAG-1 and EDAG-8) and cisplatin (concentrations of 0.5 and 1 μM). PE conjugate with an anti-Akt antibody and a flow cytometer were used in the experiment. The mean values with SD were reported based on data from three separate experiments (n = 3) conducted in triplicate. Statistical differences between the experimental groups (treated cells) and control (untreated cells) were assessed using one-way ANOVA and Dunnett’s test. *** *p* < 0.001 vs. control group.

**Figure 16 ijms-25-07764-f016:**
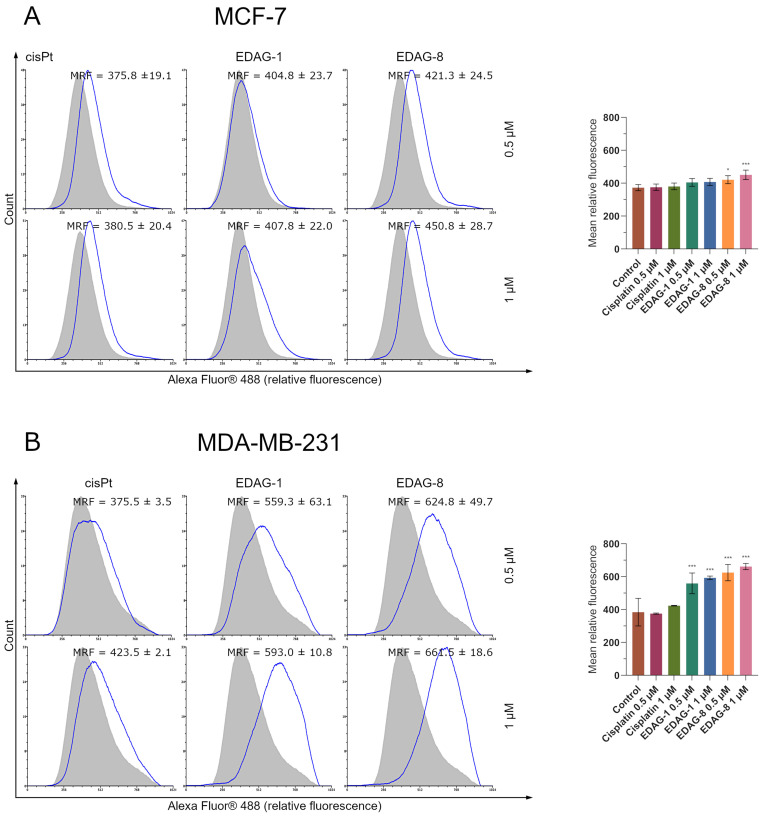
AMPK protein expression in breast cancer cells ((**A**)—MCF-7; (**B**)—MDA-MB-231) following a 24 h exposure to the tested compounds (EDAG-1 and EDAG-8) and cisplatin (concentrations of 0.5 and 1 μM). The grey histogram represents the fluorescence of the control group, and the blue colored one represents the experimental group. The fluorescence’s difference (value shift) shows the corresponding protein activity. Anti-AMPKβ1/2 primary antibody, Alexa Fluor^®^ 488 conjugate with a secondary antibody, and a flow cytometer were used in the experiment. The mean values with SD were reported based on data from three separate experiments (n = 3) conducted in triplicate. Statistical differences between the experimental groups (treated cells) and control (untreated cells) were assessed using one-way ANOVA and Dunnett’s test. * *p* < 0.05 vs. control group, *** *p* < 0.001 vs. control group.

**Figure 17 ijms-25-07764-f017:**
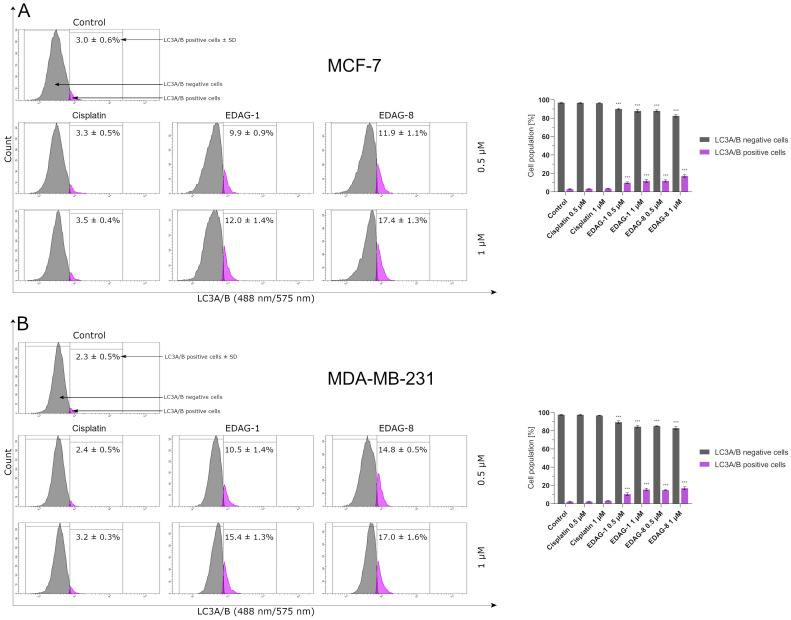
LC3A/B protein levels in breast cancer cells ((**A**)—MCF-7; (**B**)—MDA-MB-231) following a 24 h exposure to the tested compounds (EDAG-1 and EDAG-8) and cisplatin (concentrations of 0.5 and 1 μM). Alexa Fluor^®^ 488 conjugate with an anti-LC3A/B antibody and a flow cytometer were used in the experiment. The mean values with SD were reported based on data from three separate experiments (n = 3) conducted in triplicate. Statistical differences between the experimental groups (treated cells) and control (untreated cells) were assessed using one-way ANOVA and Dunnett’s test. *** *p* < 0.001 vs. control group.

**Figure 18 ijms-25-07764-f018:**
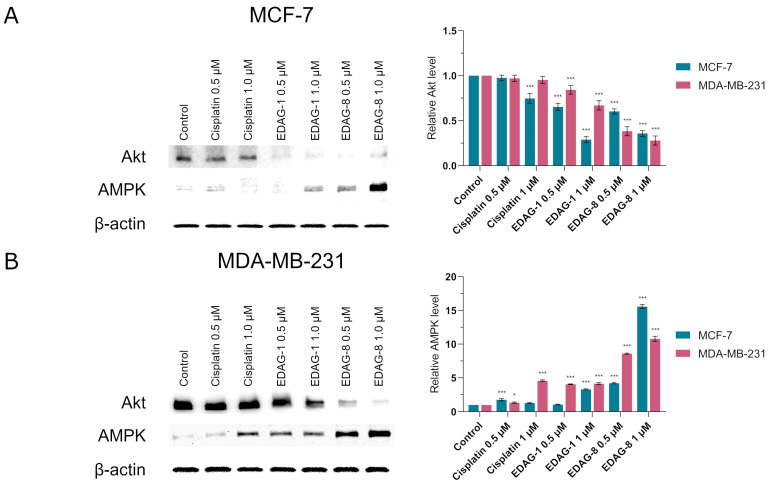
Western immunoblot analysis of the expression of Akt and AMPK proteins in breast cancer cells ((**A**)—MCF-7; (**B**)—MDA-MB-231) following a 24 h exposure to the tested compounds (EDAG-1 and EDAG-8) and cisplatin (concentrations of 0.5 and 1 μM). For the performed experiment, equal amounts (30 µg/lane) of protein lysates were used. The intensity of band staining was quantified by densitometric analysis. The mean values with SD were reported based on data from three separate experiments (n = 3) conducted in triplicate. Statistical differences between the experimental groups (treated cells) and control (untreated cells) were assessed using one-way ANOVA and Dunnett’s test. * *p* < 0.05 vs. control group, *** *p* < 0.001 vs. control group.

**Figure 19 ijms-25-07764-f019:**
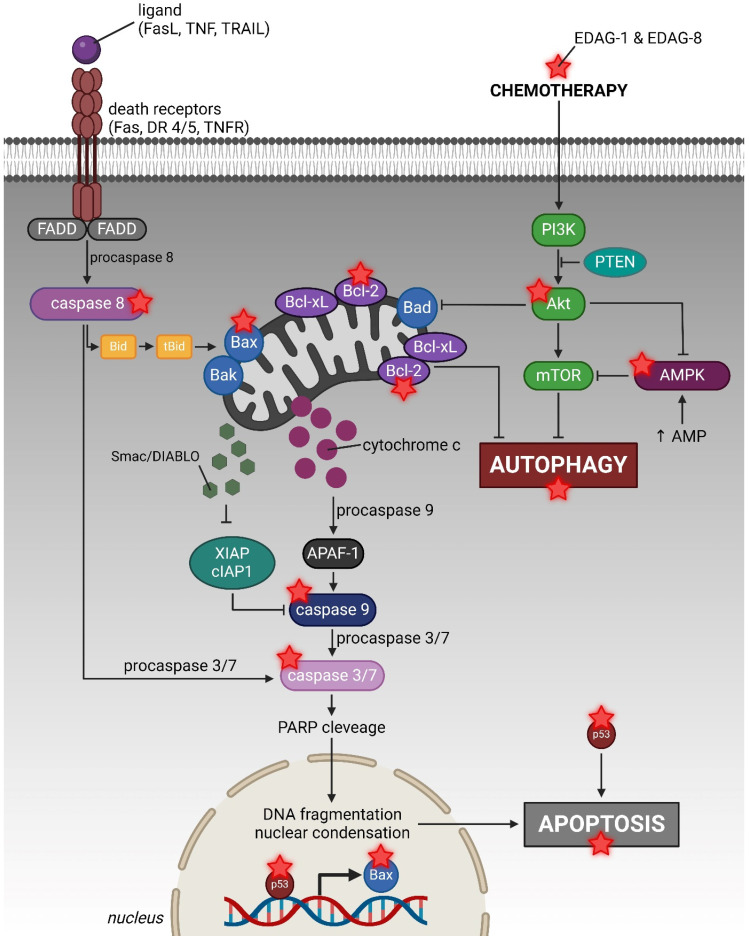
Potential molecular mode of action of the tested selenoesters EDAG-1 and EDAG-8.

**Table 1 ijms-25-07764-t001:** Characteristics of the breast cancer cell lines used in this study [44,45,46,47].

Cell Line	Cell Line Origin	Genome Ancestry	Classification	Receptor Presence	Low Level of Protein	Chemotherapy Response	HLA Class I ^a^	HLA Class II ^b^	Sequence Variations ^c^	BRCA1 Allelic Loss	BRCA1 Mutation Type	Transcript Levels of BRCA1
**MCF-7**	69-year-old Caucasian female; cells derived from pleural effusion	Mainly (56.91%) European, North	Luminal A	ER^+^, PR^+^, HER2^–^	Ki67	Often responsive	HLA-A A*02:01:01	HLA-DP DPB1*02:01:02,04:01	CDKN2A—gene deletion (9p21.3); HGNC:1787 GATA3—mutation (10p14); HGNC:4172 PIK3CA—mutation (3q26.32); HGNC:8975 TP53—mutation (17p13.1); HGNC:11998	+	Wild	Unmethylated; barely detectable
							HLA-B B*18,44	HLA-DQ DQB1*02:01,06:02				
							HLA-C C*05	HLA-DR DRB1*03,15				
**MDA-MB-231**	51-year-old Caucasian female; cells derived from pleural effusion	Mainly (60.03%) European, South	Claudin-low	ER^–^, PR^–^, HER2^–^	Ki67, E-cadherin, claudin-3, -4, and -7	Intermediate responsive	HLA-A A*02:01,02:17	HLA-DP DPB1*02:01:02,17:01	CDKN2A—gene deletion (9p21.3); HGNC:1787 CDKN2B—gene deletion (9p21.3); HGNC:1788 BRAF—mutation (7q34); HGNC:1097 KRAS—mutation (12p12.1); HGNC:6407 TERT—mutation (5p15.33); HGNC:11730 TP53—mutation (17p13.1); HGNC:11998	+	Wild	Unmethylated; normal
							HLA-B B*40:02,41:01	HLA-DQ DQB1*02:02,03:01:01				
							HLA-C C*02:02:02,17	HLA-DR DRB1*07:01,13:05				

^a^ A/B/C*—A/B/C locus. ^b^ DPB1/DQB1/DRB1*—DPB1, DQB1, DRB1 locus. ^c^ The chromosomal location of each gene mutation is given in the brackets. BRAF—B-Raf proto-oncogene; serine/threonine kinase, CDKN2A—cyclin-dependent kinase inhibitor 2A, CDKN2B—cyclin-dependent kinase inhibitor 2B, GATA—GATA binding protein 3, KRAS—KRAS proto-oncogene; GTPase, PIK3CA—phosphatidylinositol-4,5-bisphosphate 3-kinase catalytic subunit α, TERT—telomerase reverse transcriptase, TP53—tumor protein p53.

## Data Availability

Department of Synthesis and Technology of Drugs, Medical University of Bialystok, Kilinskiego 1, 15-089 Bialystok, Poland.

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
