# Peer review of "Di- and Triselenoesters—Promising Drug Candidates for the Future Therapy of Triple-Negative Breast Cancer"

_ijms, 2024, doi:10.3390/ijms25147764_

Round 1

Reviewer 1 Report

Comments and Suggestions for Authors

The present manuscript dissect the potential mechanism of action of two novel selenorganic compounds with anticancer activity. The Authors, once determined the compounds ability to inhibit cellular proliferation and DNA biosynthesis, identified apoptosis and autophagy as the molecular mechanism of action of the two compounds.

However, the manuscript is extremely long and difficult to read, with many unnecessary repetitions of concepts already mentioned and numerous list of numbers and percentage described in the text, which make the read very difficult. To this problem, I would suggest to insert the numerical data with their standard deviation within the graphs in the figures so that the main text can be more fluid. Each paragraph of the manuscript (results, discussion, material and methods) should be drastically shortened.

The entire experimental part has been carried out by flow cytometry analysis, which raises some concerns, especially in the analysis of the intracellular protein expression levels: the isotypes control are missing; most of the data are expressed as % of positive population and others as MRF: why this difference? How the MRF is calculated (without the isotype controls)?

Moreover, when analyzing intracellular protein expression levels, Western Blotting remain the main and more reliable method of analysis. For this, it is necessary to add this analysis to confirm/corroborate the results obtained by flow cytometry.

In the discussion (line 607) the sentence “we distinguish autophagy, ferroptosis, pyroptosis and necrosis, but it is apoptosis that is the main pathway…” should be revised, as the Authors only dissect autophagy and apoptosis as potential mechanism of action of the two organoesters. Therefore, they cannot exclude an involvement in ferroptosis or pyroptosis.

Author Response

Dear Reviewer, response to your review enclosed as a file.

Reviewer 2 Report

Comments and Suggestions for Authors The study of Radomska and colleague want to demonstrate the anticancer activity of new selenoester compounds. The authors have synthesized new selenium compounds with several Se atoms to test their efficacy. Even if their results are interesting in term of efficacy this work need to be improved and can’t be accepted in the present form.  After their first exeriment the authors have selected 2 compounds among 4 to characterize their anticancer activity only on MCF-7 and MDA-MB-231 cells. Since these compounds have also a strong effect on MCF-10A (normal breast epithelial cells), the authors should include these cells for all experiments conducted.  Since the authors observed strong effects after only 24h of tretament (on cell proliferation and apoptosis), they should try to find lower concentrations which could affect cancer cells and not normal cells.  To assess that compounds with several Se atoms are more efficient than selenoester with only one Se already known, they should add one of these complexes in their experiments How the authors can explain that their new selenoester compounds can induce apoptosis mediated by a DR dependant pathway?  A western blot to clearly show the cleavage of caspase 8/9, Bax and Bcl2 should be done. Similarly to be more convincing Fig 11 should include a western blot  Is this increase of p53 relevant in these cancer cells since p53 is often mutated in cancer ? All experiment performed by cytometry have to be confirmed (Idem for caspase 3/7, autophagosome, LC3, AKT, AMPK ….) the study of signaling pathway should be done by western blot to show phosphorylated and total form of proteins   These results are not enough discussed and compare to compounds already known, moreover the authors summarize in the last figure the potential molecular mecanism of action of their new selenoesters but finally they only show that these compounds can affect some pathway leading to apoptosis. it could be more appropriated to determine the target of these compounds or how they enter in cells.   

Author Response

(The authors gave the same response as above.)

Round 2

Reviewer 1 Report

Comments and Suggestions for Authors

I thank the Authors for all the effort done in answering my comments and in imroving the manuscritpt.

I have some minor comment to add.

My previuos comment on shortening the Materials and Methods section has been misinterpreted. I have seen how th Authors have done in the revised version and I don't think that cutting all the methodological part and simply citing a previous work that used the same methodology works. In comparison, the previous version of M&M was better and should be reiontegrated. I apologies for the double work.

I also suggest to add how the Authors calculated the MRF in th M&M section of those proteins expressed as MRF. 

Finally, I thank the Authors to have kept into consideration my suggestion to include Western Blot analysis to corroborate the FACS results. It would be more intuitive if the Authors could include also the densitometric analysis for the Western Blots.

Author Response

Dear Reviewer, I've attached the response to your review.
